# STUDENT-INFORMED TEACHER TRAINING

**Nico Messikommer[*], Jiaxu Xing[*], Elie Aljalbout, Davide Scaramuzza**
Robotics and Perception Group, University of Zurich, Switzerland
[*]Equal contribution

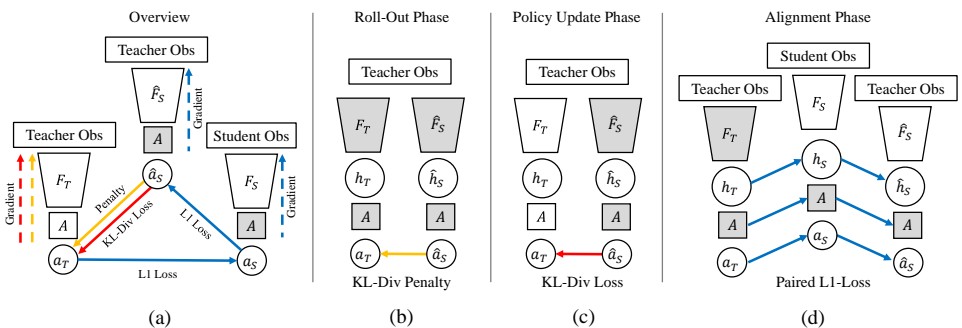

Figure 1: **Method Overview.** (a) We train three networks by freezing weights (grey box) and changing gradient flows (dashed arrow) in alternating phases. (b) In the roll-out phase, the KL-Divergence between the proxy student $\hat{F}_S$ and teacher $F_T$ is used as a penalty term. (c) Additionally to the policy gradient, the teacher encoder is updated by backpropagating through the KL-Divergence between the action distribution of the teacher and the proxy student. (d) Using student observations, the proxy student is aligned to the student $F_S$ and the student to the teacher network.

## ABSTRACT

Imitation learning with a privileged teacher has proven effective for learning complex control behaviors from high-dimensional inputs, such as images. In this framework, a teacher is trained with privileged task information, while a student tries to predict the actions of the teacher with more limited observations, e.g., in a robot navigation task, the teacher might have access to distances to nearby obstacles, while the student only receives visual observations of the scene. However, privileged imitation learning faces a key challenge: the student might be unable to imitate the teacher's behavior due to partial observability. This problem arises because the teacher is trained without considering if the student is capable of imitating the learned behavior. To address this teacher-student asymmetry, we propose a framework for joint training of the teacher and student policies, encouraging the teacher to learn behaviors that can be imitated by the student despite the latters' limited access to information and its partial observability. Based on the performance bound in imitation learning, we add (i) the approximated action difference between teacher and student as a penalty term to the reward function of the teacher, and (ii) a supervised teacher-student alignment step. We motivate our method with a maze navigation task and demonstrate its effectiveness on complex vision-based quadrotor flight and manipulation tasks.

**Multimedia Material** The project website is at https://rpg.ifi.uzh.ch/sitt/

## 1 INTRODUCTION

In reinforcement learning (RL), an agent learns to perform a task by interacting with its environment and maximizing the cumulative rewards gained through these interactions. RL has been shown to outperform human abilities in several domains (Vinyals et al., 2019; Mnih et al., 2015; Silver

This work was supported by the European Research Council (ERC) under grant agreement No. 864042 (AGILEFLIGHT).

et al., 2016; Kaufmann et al., 2023). However, this process requires extensive exploration, as the agent must avoid getting trapped in local minima, often resulting in a large number of environment interactions (Pathak et al., 2017). The number of interactions is even further increased when the agent processes high-dimensional data as input (Ota et al., 2020). Using such observations, the policy must learn to extract a notion of the agent's state, a process that is computationally expensive when optimized solely through RL. Improving the efficiency of RL training from high-dimensional observations can unlock significant advancements in various applications, particularly for robots interacting with the real world.

State-of-the-art (SotA) approaches often rely on imitation learning to accelerate training (Chi et al., 2023; Luo et al., 2024; Team et al., 2024; Doshi et al., 2024). However, collecting expert demonstrations for imitation learning can be prohibitively expensive, which has led to the development of the teacher-student framework. In this framework, expert data is generated automatically by training a teacher policy using RL on privileged task information, benefiting from efficient simulation and a faster learning process. For instance, if we take the example of autonomous driving as considered in (Chen et al., 2019), the teacher can have access to the layout of the environment, and the positions of all traffic participants as part of its observation space. In the second phase, a student policy, processing high-dimensional observations as input, is trained by using the actions of the teacher as a direct supervision signal. In the autonomous driving scenario, the student would attempt to infer the teacher's actions from visual observations. This approach eliminates the need for the student to extensively explore the environment, which can be a very challenging process when dealing with high-dimensional observations, such as images. By leveraging the direct supervision of the teacher, imitation learning significantly speeds up the training process for the student.

However, privileged imitation learning can be hindered by information asymmetry between the teacher and student, where the student receives less informative observations and struggles to imitate the behavior of the teacher (Nguyen et al., 2023). As a consequence of the information asymmetry, the teacher tends to over-rely on its full observability of the environment without considering the more limited observation space of the student. This causes the teacher to provide target actions that the student cannot infer from its observations, since the student lacks access to the same level of environmental information. For example, consider a mobile robot navigating an obstacle-filled environment. In this example, an information asymmetry in the observation space could easily arise if the teacher policy receives the relative distances to all surrounding obstacles, while the student is limited to a forward-facing camera, requiring that obstacles be within the view of the camera.

To tackle this challenge, we propose a teacher-student knowledge distillation framework that encourages the teacher to learn behaviors that account for the capabilities of the student. Specifically, the objective function of the teacher is extended by adding the upper bound of the student performance within the imitation learning setting. This results in a reward term that penalizes the teacher for visiting states where there is a significant action mismatch between the student and teacher. Additionally, minimizing this upper bound leads to a second optimization term that directly supervises the weights of the teacher network. A key feature of our approach is the dynamic prediction of the student capabilities during the teacher-environment interactions, which eliminates the need to directly render high-dimensional student observations. By embedding the upper bound of student performance for imitation learning into the learning process of the teacher, we effectively account for the teacher-student information asymmetry.

To validate our approach, we first test it in a maze setting where the teacher can choose a shortcut that is invisible to the student. Our method successfully adjusts the behavior of the teacher to take a sub-optimal route, accounting for the limited observability of the student. Additionally, we apply our framework for training a robot manipulator to open a drawer while minimizing self-occlusion in front of a camera. In this scenario, the teacher has access to the relative pose of the cabinet in addition to the internal state of the manipulator, while the student needs to compute the position of the drawer based on an image. Our method enables the teacher to modify its behavior to reduce self-occlusion, leading to a camera-aware behavior. Finally, we demonstrate the effectiveness of our method in the complex task of quadrotor flight through obstacles. The teacher, using state-based information, learns to orient the camera forward to help the student detect obstacles. Overall, our method leads to substantially higher student returns, reducing the gap between the teacher and student across tasks. As a result, our approach leads to very notable improvements in the student success rate in all considered environments.

## 2    BACKGROUND

The goal of Reinforcement Learning (RL) is to find a policy that optimizes the expected return, consisting of the accumulated and weighted reward terms obtained by trajectories governed by the underlying Markov Decision Process (MDP). The MDP is defined as a tuple $\mathcal{M} = (\mathcal{S}, \mathcal{A}, P, R, \gamma, \mu_0)$ comprising of the state space $\mathcal{S}$ and the action space $\mathcal{A}$. At the start of an episode, the agent is initialized based on the initial state distribution $d_0$ and outputs for each timestep an action $a_t \in \mathcal{A}$ sampled from the agent policy $\pi(\cdot|s)$. The transition to the state $s_{t+1}$ follows the transition probability $P(s_{t+1}|s_t, a_t)$. The performance of an agent solving the MDP can be quantified by the expected sum of rewards discounted by $\gamma$: $J(\pi) = \mathbb{E}_{s \sim \mu_0}[\sum_{t=0}^{\infty} \gamma^t r(s_t, a_t) \,|\, s_0 \sim \mu_0, a_t \sim \pi(\cdot|s_t), s_{t+1} \sim P(\cdot|s_t, a_t)]$. By iteratively interacting with the environment, RL algorithms try to find the optimal policies that maximize the expected return $\pi^* = \arg\max_\pi J(\pi)$.

In contrast, imitation learning starts already with an expert policy $\pi_T$, which is imitated by the student policy $\pi_S$. This expert policy is either directly accessible to the agent or is assumed to be previously used to collect an offline dataset of expert interactions with the environment. This setting eliminates the need for extensive exploration through environment interactions. Furthermore, a stricter supervision signal can be employed by directly providing ground truth actions from the expert to the student. However, if the data distribution for training the student policies does not represent the final application distribution, the student policy learned with imitation learning will suffer since it did not learn how to behave outside the training data distribution. In general, Xu et al. (2020); Syed & Schapire (2010); Ross et al. (2011) established an upper bound on the difference between teacher performance $J_{\pi_T}$ and student performance $J_{\pi_S}$ assuming the reward function is bounded by $r_{\max}$, i.e. $|r(s, a)| \leq r_{\max}$,

$$J(\pi_T) - J(\pi_S) \leq \frac{2\sqrt{2}r_{\max}}{(1-\gamma)^2}\sqrt{\epsilon}, \tag{1}$$

where $\epsilon$ represents the upper bound of the expected action difference between student and teacher under the discounted stationary state distribution of the expert $d_{\pi_T}(s) = (1 - \gamma)\sum_{t=0}^{\infty} \gamma^t \mathrm{Pr}(s_t = s; \pi_E)$

$$\mathbb{E}_{s \sim d_{\pi_T}}[D_{KL}(\pi_T(\cdot|s), \pi_S(\cdot|s))] \leq \epsilon. \tag{2}$$

Imitation learning is focused on minimizing this KL-Divergence by updating the student policy $\pi_S$ to predict the same action distribution as the teacher $\pi_T$.

## 3    STUDENT-INFORMED TEACHER TRAINING

As shown in Eq. 1 and Eq. 2, the performance gap between student and teacher is upper-bounded by the action difference between both policies. Thus, minimizing the action difference under the state distribution of the expert also minimizes the performance gap $J(\pi_T) - J(\pi)$. Instead of trying to minimize the action difference by adjusting the student policy $\pi_S$ (Nguyen et al., 2023; Shenfeld et al., 2023; Walsman et al., 2022), we propose to change the perspective and find a teacher policy $\pi_T$ optimizing for the task reward while considering the alignment between teacher and student. Consequently, our framework requires that the teacher and student can be jointly trained. We want to find a teacher policy $\pi_T$ that maximizes the following objective

$$\tilde{J}(\pi_T) = \mathbb{E}_{s \sim d_{\pi_T}, a \sim \pi_T(\cdot|s)}[r(s, a)] - \mathbb{E}_{s \sim d_{\pi_T}}[D_{KL}(\pi_T(\cdot|s), \pi_S(\cdot|s))] \tag{3}$$

$$= \mathbb{E}_{s \sim d_{\pi_T}, a \sim \pi_T(\cdot|s)}[r(s, a) - D_{KL}(\pi_T(\cdot|s), \pi_S(\cdot|s))] \tag{4}$$

$$\propto \mathbb{E}_{\tau \sim p_\theta}[R(\tau) - D_\theta(\tau)] = \int p_\theta(\tau)(R(\tau) - D_\theta(\tau))d\tau. \tag{5}$$

In the last step, the discounted state distribution is changed to the expectation over trajectories $\tau \sim p_\theta$, which are induced by the expert policy $\pi_T$ and represent the state and corresponding actions $\tau = \{s_0, a_0, s_1, a_1, ...\}$. Additionally, we define the return $R(\tau) = \sum_{s_t, a_t \in \tau} \gamma^t r(s_t, a_t)$ and the sum of discounted KL-Divergences $D_\theta(\tau) = \sum_{s_t \in \tau} \gamma^t D_{KL}(\pi_T(\cdot|s_t), \pi_S(\cdot|s_t))$. We use subscript $\theta$, to emphasize that the probability distribution over the trajectories $p_\theta$ and $D_\theta(\tau)$ is dependent on the parameter of the teacher network $\theta$. Following the classical policy gradient to obtain the optimal policy, we take the gradient of Eq. 5 with respect to the teacher parameters $\theta$

$$\nabla_\theta \tilde{J}(\pi_T) = \nabla_\theta \int p_\theta(\tau)(R(\tau) - D_\theta(\tau))d\tau \tag{6}$$

$$= \int \nabla_\theta p_\theta(\tau)R(\tau)d\tau - \int \nabla_\theta p_\theta(\tau)D_\theta(\tau)d\tau - \int p_\theta(\tau)\nabla_\theta D_\theta(\tau)d\tau \tag{7}$$

$$= \underbrace{\int \nabla_\theta p_\theta(\tau)(R(\tau) - D_\theta(\tau))d\tau}_{\text{Policy Gradient}} - \underbrace{\int p_\theta(\tau)\nabla_\theta D_\theta(\tau)d\tau}_{\text{KL-Div Gradient}} . \tag{8}$$

As can be observed, we end up with the standard policy gradient optimizing the task reward while also considering the teacher-student misalignment for each trajectory. This first KL-Divergence $D_\theta$ can be interpreted as a reward encouraging the teacher policy to visit states where the student and teacher are aligned and avoid states with a large misalignment. The additional "KL-Div Gradient" term contains the second KL-Divergence and represents the expectation of the gradient with respect to the teacher network over the expert states, which represents a direct supervision on the teacher weights by enforcing the prediction of the same action distribution as the student.

## 4 JOINT LEARNING FRAMEWORK

Building on the formulation in Sec. 3, we propose a practical framework to tackle the teacher-student asymmetry. Following Eq. 8, we adapt the widely-used PPO algorithm (Schulman et al., 2017) to train the teacher to learn behaviors that can be imitated by the student. At the same time, we train the student network to imitate the teacher based on a subset of the collected environment interactions containing student and teacher observations. By using a subset of paired teacher and student data, we avoid the (usually expensive) simulation of student observation for each time step the teacher interacts with the environment. An overview of the proposed method is shown in Figure 1 a) and in the Appendix A.2. In the following sections, we introduce the policy networks employed in our framework (Sec. 4.1) and detail the different training phases (Sec. 4.2).

### 4.1 ARCHITECTURE

To implement the objective in Eq. 5 inside the teacher training, we implement two key components: (i) a proxy student network taking as input teacher observations and (ii) a shared action decoder network. Excluding the critic, our method consists of three different networks: the teacher network $F_T$, the student network $F_S$, and the proxy student network $\hat{F}_S$, which all share the same action decoder network $A$.

**Proxy Student Network** To compute the action difference used in the penalty term and to obtain the KL-Div Gradient in Eq. 8, a forward pass through both the teacher and student networks is required for each collected sample. However, simulating high-dimensional student observations, such as images, is often computationally expensive, contradicting the initial goal of accelerating training. To avoid this simulation overhead, we introduce a separate neural network $\hat{F}_S$. Specifically, given the teacher observations, the proxy student $\hat{F}_S$ tries to predict the actions of the current student policy. This allows us to approximate the actions of the student at each expert state without additional simulation cost. The proxy student network is trained during the alignment phase with an L1 loss on the network activations between proxy student and student, where both student and teacher observations are available for a subset of environment interactions.

**Shared Action Decoder** Rather than using separate networks to predict actions directly from observations, we introduce shared action decoding layers used by the teacher, student, and proxy student. Each policy encodes its corresponding observations to features in a common feature space, which are then fed to the action decoder to obtain the actions. Crucially, the action decoding layers are only updated based on the policy gradient computed with the task reward. The shared action decoder allows us to leverage the high-level feature correlations learned by the teacher through extensive environment interactions without requiring student observations. As a result, both the teacher and student learn to map their observations into a common feature space instead of developing separate behavior policies. Furthermore, since the alignment between teacher and student is trained using a limited subset of data, correctly simulating the true distribution of expert states can be challenging.

In this context, the shared action decoder can help as it is trained on a broader set of expert states rather than the smaller alignment dataset, leading to more robust learning for the student.

## 4.2 TRAINING PHASES

Our proposed framework consists of three alternating training phases: (i) the classical policy roll-out, (ii) the policy update, and (iii) the alignment phase, see also Figure 1 (b)-(c). The first two phases follow the standard on-policy training, while in the alignment phase (iii), the student $F_S$ is aligned to the teacher $F_T$, and the proxy student $\hat{F}_S$ is aligned to the student $F_S$. For both network alignments, paired student and teacher observations are used. In the following, we provide more details about the specific training phases.

**Roll-out Phase** Our proposed framework introduces a minor modification to the roll-out phase of the standard teacher training, specifically in the reward computation. In addition to the task reward, we also add a penalty term computed based on the action difference between the teacher and the proxy student. This penalty encourages the teacher to only visit states in which the student can predict the same actions as the teacher, thereby improving alignment between the student and teacher. Due to the addition of this term to the reward, it influences the long-term effect of taking an action on the divergence between student and teacher. Hence, it influences the exploration behavior of the teacher. In practice, during each roll-out phase, we store a subset of expert states required in the alignment phase. Depending on the simulation envrionment, this subset can be randomly selected states or a fixed number of environments from which student observations are generated.

**Policy Update Phase** The gradient of the KL-Div term in Eq. 8 can be integrated into the policy update of the teacher, during which the network weights are updated using the clipped policy gradient of PPO. Since both the policy gradient and the KL-Divergence gradient are computed over the state distribution of the teacher, we can use the teacher states inside the roll-out buffer to compute the KL-Divergence between the action distributions of the teacher and the proxy student. This allows us to update the network parameters of the teacher in a single backward pass through the combined loss function. Note that the KL-Divergence mentioned in this phase is different to the KL-Divergence penalty introduced at the rollout phase, and is directly integrated in the loss function to push the representation of the teacher to be similar to the student. In contrast, the penalty term affects the teacher's exploration by discouraging it from visiting regions of high disagreement with the student.

In the case of a continuous action space within PPO, we can simplify the KL-Divergence further to provide intuitive insights into the KL-Divergence gradient. In this setting, the teacher actions are modeled as multivariate Gaussians of dimension $d$, with predicted mean $\mu_T$, and state-independent covariances parameters $\Sigma_T$. Consequently, the KL-Divergence term in Eq. 8 simplifies to

$$D_{KL}(\pi_T(\cdot|s_t), \pi_S(\cdot|s_t)) = \frac{1}{2}[\log \frac{|\Sigma_S|}{|\Sigma_T|} - d + \text{Tr}(\Sigma_S^{-1}\Sigma_T) \\ + (\mu_T(s_t) - \mu_S(s_t))^\intercal \Sigma_S^{-1}(\mu_T(s_t) - \mu_S(s_t))]. \tag{9}$$

The student and teacher share the same action decoder, which outputs a mean action based on the latent state it receives. For the variance, we use a state-independent parameter that is learned together with the action decoder based on teacher RL updates. Hence, the covariance matrix is the same for the student and teacher, $\Sigma_T = \Sigma_S$. As a result, the KL-Divergence reduces to the mean difference weighted by the covariance, along with an additional constant term

$$D_{KL}(\pi_T(\cdot|s_t), \pi_S(\cdot|s_t)) = \frac{1}{2}[\text{const} + (\mu_T(s_t) - \mu_S(s_t))^\intercal \Sigma_T^{-1}(\mu_T(s_t) - \mu_S(s_t))]. \tag{10}$$

By optimizing Eq. 10, the teacher network $F_T$ is aligned to the proxy student network $\hat{F}_S$. Intuitively, the loss increases when the teacher is confident in its actions (i.e. low $\Sigma_T$), but there is still significant misalignment between the teacher and student. Consequently, the gradient update increases the covariance, leading to greater exploration during the roll-out phase, which increases the likelihood of the teacher discovering behaviors that are feasible for the student to learn.

**Alignment Phase** The alignment phase focuses on aligning the features across the encoders of the teacher, proxy student, and student. This phase is the only one that requires paired teacher and student observations, which are simulated from a subset of the teacher's experiences during the roll-out phase. We align the student encoder with the teacher encoder by computing the L1 loss between

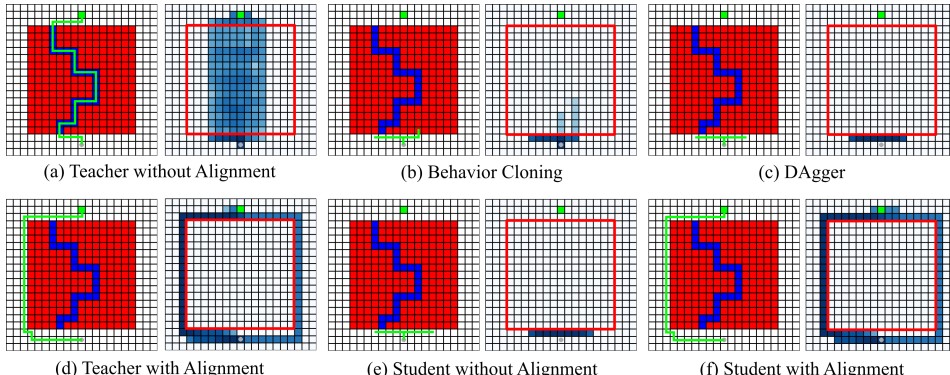

Figure 2: The goal of the agent is to navigate from the start (grey point) to the goal (green cell). The environment consists of four types of cells: empty (white), lava (red), and path (blue). The teacher can see all cell types while the student can not distinguish between lava and path. A teacher trained without alignment finds its optimal path through the maze (a), which can not be imitated by the student trained without alignment (e), Behavior Cloning (b), and DAgger (c). In contrast, a teacher trained with alignment navigates around the maze (d), which can be easily copied by the student (f).

their corresponding features and between the activations of the frozen shared action decoder. To prevent the collapse of the model into predicting constant outputs, gradients are only backpropagated to the student encoder. Similarly, the proxy student is aligned with the student using the L1 loss on the encoded features, with gradients only backpropagated to the proxy student. The parameters of the teacher remain unaffected during this phase and are only updated during the policy update phase.

## 5 EXPERIMENTS

We compare our method to multiple baselines introduced below. We evaluate our student-informed teacher training framework on three diverse tasks: maze navigation in a tabular setting, vision-based obstacle avoidance with a quadrotor, and vision-based drawer opening using a robot arm. Finally, we perform ablations to understand the role of different aspects of our method. For more details about the experimental setup and training, we refer to the Appendix Section A.1.

**Baselines** We compare our method against multiple Imitation Learning baselines, which were trained with the same number of rendered images and environment interactions, including teacher training. Furthermore, all baselines use the same network architecture for the student and teacher.

- **Behavior Cloning (BC)** The dataset, comprising privileged state information and rendered images, is generated from the fixed number of timesteps of the teacher. The student network is then trained by minimizing the action loss between student and teacher on the collected data.

- **DAgger** (Ross et al., 2011) We employ an exponentially decreasing sampling threshold ($\beta_0 = 0.98$) to decide whether the actions of the teacher or student are used at each timestep. Similarly to BC, we minimize the loss between the student and teacher actions on the collected samples.

- **HLRL** (Radosavovic et al., 2024) The weighting between the behaviour cloning loss and the vision-based RL loss is epxonentially decreased ($\beta_0 = 0.9998$). To achieve successful runs, a shared action decoder was necessary.

- **DHBC** (Fu et al., 2022) We treat the *Privilged Info Encoder* as the teacher and the *Adaptation Module* as the student encoder, with the *Unified Policy* representing the shared action decoder. Both encoders are updated using their dual L1-loss in the feature space.

- **COSIL** (Nguyen et al., 2023) We adapt the method to PPO by using the L1 loss between student and teacher in the reward. As proposed in (Nguyen et al., 2023), the RL and IL objective is weighted with a learnable parameter, which is updated based on a target distance (0.5).

- **w/o Alignment** (Ours) We exclude the KL-Divergence from both the reward and the policy update phase while keeping the paired L1-Loss on shared action decoder features. As a result, the student does not affect the teacher.

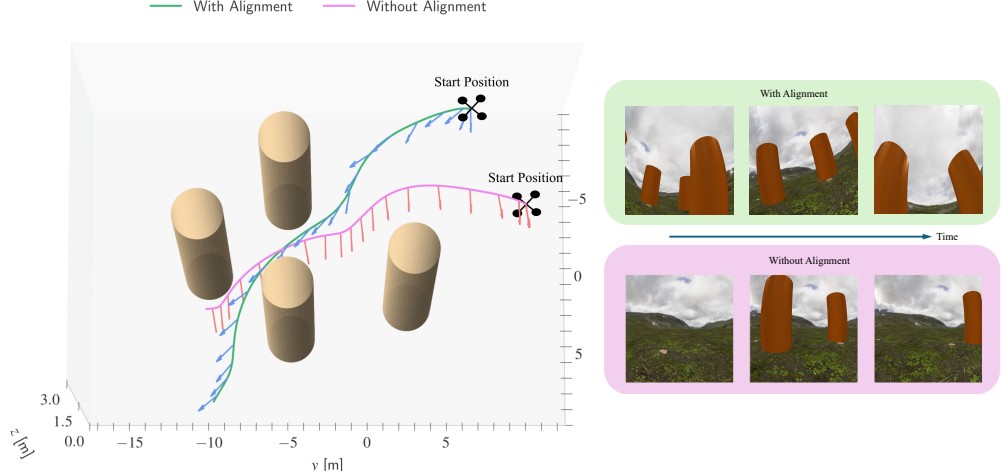

Figure 3: **Vision-Based Obstacle Avoidance.** On the left, the teacher trajectory rollouts of the vision-based quadrotor obstacle avoidance tasks are visualized. Our approach results in a policy behavior where the quadrotor adjusts the camera's viewing direction to capture sufficient environmental information for the student policy.

**Implementation Details** For the vision-based tasks, we use a three-layer MLP with ELU activations for the teacher and the proxy student, which both receive privileged observations. The shared action decoder consists of a single fully connected layer. The student processes images using a frozen DINOv2 encoder (Oquab et al., 2023). These image features are combined with state observations and passed through an additional MLP before being forwarded to the shared action decoder.

## 5.1 ENVIRONMENTS

**Color Maze** We first test our method on a maze navigation task in a tabular setting. The objective of the agent is to reach the goal (green cell) from the starting point (grey point), as visualized in Figure 2. In the center of the environment is a maze where the agent can only navigate along a randomly generated path (blue cells), and the episode ends if the agent steps into the lava (red cells) or reaches the target. The teacher observations include the distance to the goal, the type of neighboring cells (empty, lava, or path), and the relative movement from the previous timestep. The student receives the same observations, except that the cell types are classified as empty or occupied, which maps lava and path cells to the same value, which leads to a teacher-student asymmetry. The reward structure assigns a positive reward for reaching the target (10) and for visiting each new path cell (0.5), while penalties are given for moving into lava (0.1) and revisiting path cells (0.5).

**Vision-based Obstacle Avoidance with a Quadrotor** We evaluate our approach on the complex task of agile quadrotor control, in which a quadrotor needs to follow a constant velocity command while avoiding obstacles. To simulate this, we design a custom environment using the realistic agile quadrotor simulator, Flightmare (Song et al., 2021). The environment is a world box measuring $30\text{m} \times 30\text{m} \times 3\text{m}$, within which we randomly place four static pillar obstacles, each with a radius of 1.5m. At the start of each episode, the quadrotor's position and viewing direction are randomly initialized at collision-free locations. Next, we sample a velocity command for the quadrotor that would result in a potential collision with one of the obstacles. The teacher policy receives observations composed of the quadrotor's linear velocity, commanded linear velocity, orientation, angular velocity, and the relative distance to each obstacle. In contrast, the vision-based student policy replaces obstacle positions with inputs from an RGB camera with a limited field of view, as shown in Figure 3. The reward is detailed in appendix A.1.2. During our experiments, we conduct 256 runs with random initialization, each running for 1000 steps for all the baseline approaches.

**Vision-based Manipulation** We further evaluate our method on a complex vision-based manipulation scenario, where a robot arm learns to open a drawer. We adapt the publicly available *Omniverse Isaac Gym Reinforcement Learning Environments for Isaac Sim* repository, specifically modifying the cabinet-opening task. In this task, a Franka robot arm is trained to open a drawer that contains objects inside. The teacher receives observations composed of the robot's joint positions and velocities and the relative position between the gripper and the drawer handle. The student observations

| Methods | Success Rate |
|---|---|
| BC | $0.05 \pm 0.04$ |
| DAgger | $0.08 \pm 0.03$ |
| HLRL | $0.31 \pm 0.11$ |
| DWBC | $0.35 \pm 0.07$ |
| COSIL | $0.30 \pm 0.07$ |
| w/o Align (Ours) | $0.38 \pm 0.11$ |
| w Align (Ours) | $\mathbf{0.46 \pm 0.04}$ |

Table 1: **Obstacle Avoidance Success Rates.** The mean and standard deviation of the success rate for vision-based quadrotor flight obtained from three trainings.

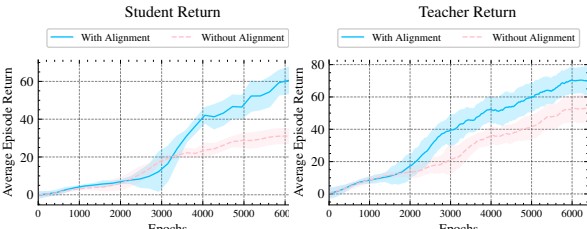

Figure 4: **Obstacle Avoidance Returns.** The mean returns achieved by the student and teacher trained with and without alignment, averaged over three training runs.

replace relative distance information with an image. Instead of an unobstructed top view, the viewpoint of the camera is selected closer to the robot arm. This camera setup is closer to real-world platforms, such as humanoid robots, where certain arm configurations may obstruct cameras.

## 5.2 RESULTS

**Color Maze** In Figure 2, we show a single trajectory (green path) and the occupancy grid (blue cells) for 15 evaluation environments across five training runs for each tested policy. As can be seen in Figure 2, students trained with Behavior Cloning (BC) or DAgger fail to imitate the behavior of the teacher due to their inability to differentiate between lava and path cells. In Figure 2 (e), we observe a similar behavior for the student trained with a shared action decoder but without teacher alignment. In contrast, when using our framework, with a teacher penalty for visiting states where the student is unable to predict the same action, i.e., inside the maze, the teacher learns to avoid the maze and navigates around to reach the target. This behavior is successfully imitated by the student, who reaches the target in all test runs, unlike all other baselines. These results confirm that our framework helps the teacher learn strategies that are easier for the student to imitate, such as avoiding the maze altogether. Additionally, since the behavior of navigating around the maze leads to sparse rewards once the goal is reached, the results demonstrate that the penalty term does not hinder the exploration necessary to discover the optimal student solution.

**Vision-based Obstacle Avoidance with a Quadrotor** The results shown in Table 1 clearly show that our approach with alignment achieves the highest success rate with 0.46 compared to 0.08 (DAgger) and 0.05 (BC). Moreover, we show that our approach surpasses another baseline approach DWBC by 0.11, HLRL by 0.15, and COSIL by 0.16, emphasizing that our adaptive teacher training offers more effective guidance to the student than the integration of RL rewards and IL with a static teacher. Table A.1.3 highlights the perception awareness metrics, showing our approach improves the student policy's behavior and performance. This is due to the fact that, the teacher policy lacks "perception awareness of obstacles," meaning the obstacles are not fully observable during the avoidance task from the equipped camera. As a result, the vision-based student struggles to infer the correct actions, leading to very low success rates (below 0.1), despite all teacher policies achieving over 98% success rates. This highlights the importance of enforcing information symmetry between teacher and student policies. The reason the success rates for the imitation baselines are not zero is that, in some configurations, the quadrotor is initialized with a viewing angle that allows the student to detect obstacles, enabling it to replicate the teacher's policy. Figure 3 illustrates the rollout trajectories, where we observe that the policy learned through our approach adjusts the viewing direction during flight to align with the velocity direction. This ensures that all encountered obstacles are visible in the camera's field of view, allowing the student to infer the necessary actions using sufficient information. Figure 4 shows the returns of our method with and without alignment for both the teacher and student policies. The results show that our proposed alignment is crucial for the student to reach reward regions similar to the teacher's despite the student's more limited observability.

**Vision-based Manipulation** Table 3 reports the success rates of the student for our method (with and without alignment) and the baselines across five training runs. Our framework with and without alignment significantly outperforms students trained with DAgger and BC, with success rates of up to 0.77 compared to 0.47 (DAgger) and 0.27 (BC). This improvement can be explained by the

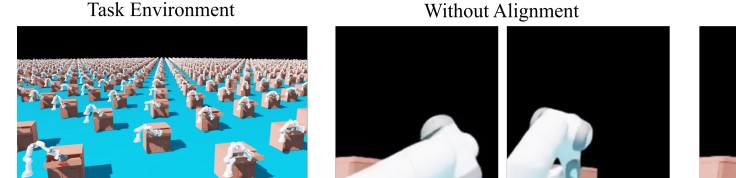

Figure 5: **Vision-Based Manipulation.** On the left, the task of opening a drawer with a robotic arm is visualized for all of the parallel environments. The two images in the center (Without Alignment) are sample images given to the student, which show the teacher behaviors trained without our alignment. Our approach with alignment leads to behaviors that the student can imitate more easily, i.e., the robot does not block the red drawer handle, as visualized in the two images on the right.

shared task decoder, which is trained by leveraging the training samples of the teacher. Additionally, our method outperforms DWBC, HLRL, and COSIL with a success rate improvement in the range of 0.25-0.32, which confirms that the adaptation of the teacher provides better student supervision than the combination between RL reward and IL with a fixed teacher. Our method with alignment improves student success rates by 17% compared to the non-aligned framework while also consistently achieving higher returns (Figure 6). These results demonstrate that our framework helps the teacher learn better behaviors for the student. Student policies trained without alignment achieve non-zero success rates due to the small sampling interval of the cabinet position. This allows them to memorize behaviors without relying heavily on the images. All teachers, regardless of alignment, reach a 100% success rate. Interestingly, the return of the teacher trained with alignment is also constantly higher than without alignment. A possible explanation is that the teacher learns gripper movements optimized for robustness without trying multiple times to grab the handle, which is a difficult behavior for the student lacking relative pose information.

As can be seen in Figure 5, our method leads to several different behaviors. With alignment, the teacher learns once to grab the handle from a top-down configuration while another teacher lowers its first two links to make the red handle visible. In both cases, the red handle is visible right before the gripper touches it. This shows that our alignment leads to emerging teacher behaviors that consider the imitation difficulties of the student.

**Ablations** To validate our design choices, we perform multiple ablations on the manipulation environment, reported in Table 2. Namely, we study the effect of the penalty term (w Penalty), the KL-Divergence gradient in the policy update (w KL-Grad), and the role of the shared action decoder. Our results indicate that the KL-Divergence gradient and the shared action decoder are crucial to the large improvement achieved by our method. As for the alignment penalty, its absence leads to a 14% drop in performance. Finally, we study the sensitivity to the coefficient $\lambda_1$ of the alignment penalty. Our results show clear robustness to this choice, as our method consistently achieves a high success rate regardless of the choice. Setting $\lambda_1 = 0.025$ leads to an impressive 95% success rate.

## 6 RELATED WORK

Imitation learning from a teacher policy has shown significant progress across various domains. Approaches such as behavior cloning (BC) (Bain & Sammut, 1995) and DAgger (Ross et al., 2011) typically train on a dataset of encountered states paired with expert actions. The core idea is to train a student policy to imitate the behavior of either a reinforcement learning (RL) policy that has been trained using state information or through expert demonstrations. In BC (Bain & Sammut, 1995), the objective is to learn a policy that closely matches expert demonstrations through supervised learning. DAgger (Ross et al., 2011) improves upon this by aggregating expert demonstrations with the student policy's experiences during training to enhance generalization beyond the original demonstrations. Ross & Bagnell (2014) extended DAgger by taking into account not only the expert's actions but also the long-term consequences of the student's policy quantified using the task cost. Ho & Ermon (2016) propose a method for matching the state-action distribution using a generative adversarial algorithm similar to the one used in generative adversarial networks (Goodfellow et al., 2014). Other methods explored leveraging priviliged information in components that are only relevant for training such as critics, reward models and dynamics models (Pinto et al., 2018; Hu et al., 2024).

One key assumption shared by these methods is that the observation information available to the student policy must be at least as comprehensive as that available to the teacher (Osa et al., 2018;

| | w Penalty / w/o KL-Grad | w/o Penalty / w KL-Grad | w/o shared decoder | $\lambda_1 = 0.1$ | $\lambda_1 = 0.075$ | (Default) $\lambda_1 = 0.05$ | $\lambda_1 = 0.025$ |
|---|---|---|---|---|---|---|---|
| Success Rate | $0.47 \pm 0.37$ | $0.74 \pm 0.08$ | $0.62 \pm 0.27$ | $0.81 \pm 0.20$ | $0.77 \pm 0.18$ | $0.88 \pm 0.07$ | $0.95 \pm 0.03$ |

Table 2: **Ablations for Vision-Based Manipulation.** The mean and standard deviation of the success rate obtained from five trainings runs.

| Methods | Success Rate |
|---|---|
| BC | $0.16 \pm 0.15$ |
| DAgger | $0.34 \pm 0.31$ |
| HLRL | $0.61 \pm 0.22$ |
| DWBC | $0.63 \pm 0.18$ |
| COSIL | $0.56 \pm 0.21$ |
| w/o Align (Ours) | $0.61 \pm 0.18$ |
| w Align (Ours) | **$0.88 \pm 0.07$** |

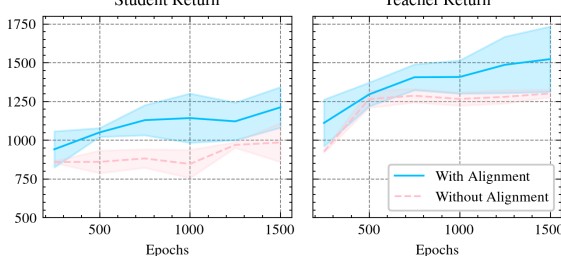

Table 3: **Manipulation Success Rates.** The mean and standard deviation of the success rate for the task of opening the drawer obtained from five trainings runs.

Figure 6: **Manipulation Returns.** Mean returns of the student and teacher trained with, without alignment and DWBC, averaged over five training runs.

Swamy et al., 2022). This assumptions falls short in many scenarios, especially when the student operates in a partially observable setting. In such cases, the student could easily struggle to infer the teacher's behavior based on its own partial observability. Hence, multiple methods proposed to augment the teacher distillation objective with a student RL objective (Weihs et al., 2021; Nguyen et al., 2023; Shenfeld et al., 2023; Radosavovic et al., 2024). These methods introduce different ways to balance the imitation and RL terms, either using fixed trade-off coefficients (Nguyen et al., 2023), a fixed schedule (Radosavovic et al., 2024), or some elaborate heuristic (Weihs et al., 2021; Shenfeld et al., 2023). Similarly, Walsman et al. (2022) jointly train two separate policies besides the teacher: a follower, which learns to follow the teacher, and an explorer, which maximizes environmental rewards using the follower's value function for reward shaping. Alternatively, in an off-policy setting, the agent can directly optimize the student policy with RL using interactions collected by rolling out the teacher to facilitate exploration (Chane-Sane et al., 2024). In contrast to these methods which focus on guiding the student exploration, we approach the challenge of asymmetric learning by regularizing the teacher policy. A similar concept have been explored for multi-agent self-play (Hamade et al., 2024) and student-teacher training (Warrington et al., 2021; Fu et al., 2022). Namely, Warrington et al. (2021) propose exposing the teacher training to student rollouts to ground the behaviors learnt by the teacher. Fu et al. (2022); He et al. (2024) learn a unified student-teacher policy and two separate encoders which are encouraged to extract similar features. In the context of generative models, Bachman & Precup (2015) proposes to jointly optimize a primary policy and a guide policy to reconstruct image regions. Our approach augments the teacher loss by adding a loss that penalizes confident actions in areas where the teacher-student discrepancy is high. Additionally, we modify the teacher's reward to discourage visiting states where the student struggles to infer teacher actions.

## 7 CONCLUSION

This paper tackles the problem of teacher-student asymmetry in imitation learning. To address this problem, we propose a novel method that trains the teacher not only based on task objectives but also by encouraging behaviors that the student can successfully imitate. By including the performance bound in imitation learning in the teacher objective, we derive two key modifications to the teacher training: a reward term penalizing states with a high teacher-student misalignment and directly aligning the teacher to the student based on the KL-Divergence between their action distributions. Our approach is not limited to specific IL algorithms and can seamlessly integrate with various IL pipelines, including multi-agent IL. We illustrate our method in a tabular maze task, where it outperforms standard imitation learning baselines that fail to reach the target. Furthermore, our framework enhances student performance in the complex task of vision-based quadrotor obstacle avoidance, where perception-aware flight emerges without the need for explicit reward tuning. We also validate our method in a vision-based manipulation task, showing how the teacher learns to open a drawer without obstructing critical visual information in the camera view of the student. We believe that our framework extends privileged imitation learning to various modalities with large information gaps, unlocking the potential to tackle complex real-world tasks in robotics.

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

# A APPENDIX

## A.1 TRAINING DETAILS

### A.1.1 COLOR MAZE

We implemented the color maze environment using the Gym framework (Brockman et al., 2016), with the discrete agent trained via the PPO (Schulman et al., 2017) implementation of StableBase-lines3 (Raffin et al., 2021) based on Pytorch (Paszke et al., 2017). The training setup involves 1,000 parallel environments, each containing a randomly generated path through the maze located at the center of a 21x21 grid. The maze itself spans 15x15 cells, leaving a three-cell-wide empty border around it, through which the optimal path of the student goes. The paths are generated randomly by alternating between vertical segments of 3 to 5 cells and horizontal segments of 1 to 4 cells. Given the large number of environments, each maze path is generated only once at the start of training and remains fixed throughout the process to save computation time.

The action space of the agent consists of four discrete movements: one cell up, right, down, or left. If the agent selects an action that would move it outside the grid, its position remains unchanged. Moving into a lava cell results in immediate episode termination and a negative reward. In contrast, reaching the target cell terminates the episode with a positive terminal reward.

The teacher training uses in total $10^8$ environment interactions. During each roll-out phase, 250 steps are collected from each of the 1,000 parallel environments, resulting in 250,000 experiences. In the policy update phase, the teacher network is updated using a batch containing all 250,000 col-lected experiences. In addition to the default PPO gradients, we apply the KL-Divergence gradient

with a weighting coefficient of 0.001. The entropy coefficient for PPO is set to 0.3. During each paired alignment step, the observations stored in the roll-out buffer are used as a single batch to update the imitated student and student networks over 20 iterations. The networks are optimized using the L1 loss between corresponding features, with network weights updated via the ADAM optimizer (Kingma & Ba, 2015).

The reward function consists of terminal rewards for either reaching the target ($r_{success}$) or stepping into a lava cell ($r_{fail}$). To encourage the teacher to follow the correct path through the maze, we assign a reward when the agent moves to a path cell for the first time ($r_{path}$). To prevent the teacher from simply cycling over already-visited path cells, we impose a penalty for revisiting the same path cell $r_{revisit\_path}$. In the case of our alignment framework, an additional penalty term based on the KL-Divergence between teacher and student actions is included in the reward function $r_{KL}$.

$$r_{tot} = 10r_{success} + 0.5r_{path} - 0.5r_{revisit\_path} - 0.1r_{fail} - 1.9r_{KL}. \tag{11}$$

### A.1.2 VISION-BASED QUADROTOR FLIGHT

For the obstacle avoidance task, we build a customized RL training environment using Flightmare and Stable-baselines3. In this task, we define the observations as $\boldsymbol{o}_{\text{obstacle}} = \left[\tilde{\boldsymbol{R}}, \boldsymbol{v}_{\text{cmd}}, \boldsymbol{v}, \boldsymbol{\omega}, a_{\text{prev}}, \delta\boldsymbol{p}_1, \delta\boldsymbol{p}_2, \delta\boldsymbol{p}_3, \delta\boldsymbol{p}_4\right]$, where $\tilde{\boldsymbol{R}} \in \mathbb{R}^6$ is a vector comprising the first two columns of $\boldsymbol{R}$, $\boldsymbol{v} \in \mathbb{R}^3$ and $\boldsymbol{\omega} \in \mathbb{R}^3$ denote the linear and angular velocity of the drone, $a_{\text{prev}}$ represents the previous action from the actor policy, and $\delta\boldsymbol{p}_1, \delta\boldsymbol{p}_2 \in \mathbb{R}^{12}$ represent the relative difference in position of Here, $\delta\boldsymbol{p}_i$ represents the absolute distances of the quadrotor to all obstacles represented in the world frame. The total reward at time $t$, denoted as $r_t$, consists of several components:

$$r_{\text{obstacle}} = r_{\text{prog}} + r_{\text{act}} + r_{\text{br}} + r_{\text{obstacle crash}} + r_{\text{crash}}, \tag{12}$$

where $r_{\text{prog}}$ represents progress toward command velocity direction to ensure velocity following, $r_{\text{act}}$ penalizes changes in actions from the previous time step, $r_{\text{br}}$ discourages high body rates to ensure stable flying behavior, $r_{\text{pass}}$ is a penalty for crashing to encountering obstacles, weighted by the velocity. and $r_{\text{crash}}$ is a crashing penalty when it collides to the ground or flies out of the world box. The policy is trained using standard PPO implemented in the stable-baselines library. The reward components are formulated as follows

$$\begin{aligned}
r_{\text{prog}} &= 2.0 \sum_i \delta\boldsymbol{p}_i, \\
r_{\text{act}} &= 0.15\|a_t - a_{t-1}\|, \\
r_{\text{br}} &= 0.05\|\boldsymbol{\omega}_{\mathcal{B},t}\|, \\
r_{\text{obstacle crash}} &= -10\|\boldsymbol{v}\| \quad \text{if robot crash to the obstacle,} \\
r_{\text{crash}} &= -5\|\boldsymbol{v}\| \quad \text{if robot crashes (ceiling, ground).}
\end{aligned} \tag{13}$$

Over the course of training, the teacher collects data across 6300 epochs, with 100 parallel environments. Among these environments, 64 render images at each timestep. During the policy update phase, we use a minibatch size of 12500.

### A.1.3 VISION-BASED MANIPULATION

For the manipulation task, we use the Omniverse Isaac Gym Environments framework. Specifically, we modify the existing Franka Cabinet Task, where a Franka arm is tasked with opening the top drawer of a cabinet. We make several minor modifications to this setup. First, we change the color of the cabinet to brown and the top drawer handle to red to improve visibility and highlight when the handle is occluded. Second, we remove the knobs from the bottom doors and the handle from the bottom drawer to eliminate alternative visual cues for locating the top handle. To increase occlusion, we raise the Franka arm by 0.4m and rotate the cabinet (60 degrees around the z-axis), as shown in Fig. 5. Additionally, we uniformly sample the x and y positions of the cabinet relative to the robot arm within a grid of [-0.15m, 0.15m] × [-0.1m, 0.1m]. The initial position of the robot arm was chosen to ensure that the default teacher consistently occludes the camera while still maintaining enough distance between the robot arm and the drawer handle to ensure a certain degree of exploration.

The teacher agent is trained using the RL-Games framework (Makoviichuk & Makoviychuk, 2021), specifically with a PPO continuous agent. Over the course of training, the teacher collects data across 1,500 epochs, with a horizon of 16 timesteps and 4,096 parallel environments, resulting in a total of 98,304,000 environment interactions. Among these environments, 128 render images at each timestep, generating 3,072,000 data samples for the paired alignment phase. To manage these image samples, we employ a rolling buffer with a size of 100,000, updating it in a FIFO (First In, First Out) manner. During the policy update phase, we use a minibatch size of 8,192. The KL-Divergence loss is weighted by 0.01 in the policy update. For all other PPO hyperparameters, we use the default settings provided by the Omniverse Isaac Gym framework.

The action space of the robot arm is the multi-step integration joint position Aljalbout et al. (2024). The Panda arm has 7 joints and the gripper has 2 fingers, making for a total of 9 degrees of freedom. For the teacher, we use the default observation set, which includes the joint positions, joint velocities, the relative 3D distance between the gripper and the drawer handle, as well as the 3D position and velocity of the drawer joint. In contrast, the student receives only the joint positions and angular velocities, along with an additional image input, as shown in Fig. 5. We use the default reward formulation, which includes components such as the distance between the gripper and the handle, the gripper's orientation, and the position of the gripper's "thumbs," among others. For our alignment training, the KL-Divergence is weighted by 0.05 and added to the task reward.

## A.2 Algorithm Overview

A pseudocode description of the different training phases with the crucial steps is provided in Algorithm 1.

---

**Algorithm 1:** Student-Informed Teacher Training

---

**Input:** Teacher policy $\pi_T$, Student policy $\pi_S$, Proxy student policy $\hat{\pi}_S$, Task environment $\mathcal{E}$, Reward function $r$, KL-divergence weight $\lambda_1, \lambda_2$, Iterations $N$, Roll-out steps $T$, Policy update steps $M$, Alignment steps $L$

**Output:** Trained teacher $\pi_T$ and student $\pi_S$

**Initialize:** Teacher, student, proxy encoders, and shared task decoder

**Buffers:** Experience buffer $\mathcal{B}_{\text{exp}}$, Alignment buffer $\mathcal{B}_{\text{align}}$

**for** $i \leftarrow 1$ **to** $N$ **do**
    // Roll-Out Phase
    **for** $t \leftarrow 1$ **to** $T$ **do**
        Sample action $a_t \sim \pi_T(s_t)$
        Execute $a_t$ in $\mathcal{E}$ to get $r_t, s_{t+1}$
        Update reward: $r'_t \leftarrow r_t - \lambda_1 D_{KL}(\pi_T(s_t) \| \hat{\pi}_S(s_t))$
        Add $(s_t, a_t, r'_t, s_{t+1})$ to $\mathcal{B}_{\text{exp}}$
        Render $o_{\text{stud}}$ for a subset of visited states and add to $\mathcal{B}_{\text{align}}$
    **end**
    // Policy Update Phase
    **for** $j \leftarrow 1$ **to** $M$ **do**
        Sample experiences from $\mathcal{B}_{\text{exp}}$
        Compute loss: $\mathcal{L} = \mathcal{L}_{\text{policy}} + \lambda_2 D_{KL}(\pi_T \| \hat{\pi}_S)$
        Update teacher policy $\pi_T$ using $\mathcal{L}$
    **end**
    // Alignment Phase
    **for** $k \leftarrow 1$ **to** $L$ **do**
        Sample $(o_T, o_S) \sim \mathcal{B}_{\text{align}}$
        Update $\pi_S$ by aligning teacher and student features
        Update $\hat{\pi}_S$ by aligning student and proxy features
    **end**
**end**

---

### A.3 EVALUATION ON LEARNING PERCEPTION AWARENESS

To further demonstrate the effectiveness of our approach, we evaluated how the resulting policies are essentially perception-aware. In our experiments, we argue that the vision-based student policy in our framework can learn to fly in a manner that effectively perceives the environment using the onboard camera. To validate this, we used two different metrics: the *Velocity Angle* and *Number of Obstacles in View*. The *Velocity Angle* measures the angle between the quadrotor's heading (camera viewing direction) and its velocity direction. Ideally, the quadrotor should "look at" the direction it is flying towards. A smaller angle ensures that obstacles are perceived in time, allowing the controller to gather sufficient information to act effectively. Additionally, we introduced another direct metric, the *Number of Obstacles in View*, which quantifies how many obstacles the policy can perceive during each rollout. As shown in A.1.3, our approach demonstrates superior performance in terms of perception awareness compared to all baseline methods. This further underscores the effectiveness of our approach.

| Methods | Velocity Angle [deg] ↓ | Num. Obstacle in View ↑ |
|---|---|---|
| BC | 75.5 | 1.92 |
| DAgger | 78.6 | 2.42 |
| HLRL | 61.9 | 2.09 |
| DWBC | 46.7 | 2.33 |
| COSIL | 69.3 | 2.24 |
| w/o Align (Ours) | 63.2 | 2.61 |
| w Align (Ours) | **32.2** | **3.51** |

