# OpenReview forum: "Student-Informed Teacher Training"
_ICLR.cc/2025/Conference — ICLR 2025 Spotlight_

### Official Review · Reviewer_Xbsv · 2024-11-02

**Soundness:** 2
**Presentation:** 2
**Contribution:** 3
**Rating:** 6
**Confidence:** 4

**Summary:**

To address the asymmetry in observation spaces during teacher-student training in reinforcement learning, this paper introduces a KL divergence penalty term between the teacher policy and a proxy for the student policy. This penalty is added to the teacher's RL loss, encouraging the teacher to adapt its behavior based on the student's observation space. To facilitate efficient penalty computation during teacher training, the authors employ a proxy student with the same privileged observation space as the teacher. The student learns by imitating the teacher from its limited observation space, while the proxy student imitates the student using privileged observations. They validate their approach in simulation across a toy grid navigation task, a vision-based quadrotor obstacle avoidance environment, and a vision-based manipulation environment, demonstrating the emergence of perception-aware behaviors.

**Strengths:**

- Impressive qualitative results: The emergence of perception-aware behaviors is impressive, and the authors effectively demonstrate that their approach produces this outcome across different environments, including a quadrotor flight task and a vision-based manipulation task.
- Interesting approach: The teacher-student observation asymmetry is a relevant problem to address in robotics. The idea of penalizing the teacher for behaviors that cannot be reproduced from the student’s observation space is both simple and sound.

**Weaknesses:**

- Missing important comparison and discussion: Several prior works focus on training the student with RL to achieve behaviors distinct from the teacher [1, 2, 3, 4, 5, 6, 7] (and likely more). While some of these are referenced in the paper, there is no thorough discussion of this class of methods, and the authors only compare their approach to BC and DAgger, which were not specifically designed to handle observation asymmetry. Since many of these prior works are motivated by teacher-student observation asymmetry, it is crucial that the authors include a proper discussion of this line of work and compare their approach to some of them.
- Clarity and methodological details: Certain aspects of the approach, especially in Section 4, lack clarity. The method involves alternating between data collection, RL policy updates with the proposed penalty, and policy alignment, but critical details are reserved for the appendix, such as the ratio of privileged to high-dimensional observation samples or the number of updates performed in each phase for each component. A potential drawback of this approach is that alternating training for both teacher and student before teacher training is complete might demand more high-dimensional samples than BC or DAgger, which train the teacher once and then the student. It is unclear how comparisons to baselines were made, particularly regarding environment interactions. Does the proposed method require more interactions than the baselines? Were baselines evaluated with equivalent budgets of privileged and high-dimensional samples? Additionally, compared to the baselines, the approach introduces a proxy student, a more involved training protocol and a KL penalty with associated hyperparameters, making it challenging to assess the algorithm's overall complexity. The “w/o Align” baseline is also not clearly explained, nor is it clear why its behavior differs from BC and DAgger. Finally, the shared action decoder is introduced without sufficient motivation; the authors could clarify this design choice, for instance through ablation studies.

References:
- [1] TGRL: An Algorithm for Teacher Guided Reinforcement Learning, ICML2023
- [2] Leveraging Fully Observable Policies for Learning under Partial Observability, CoRL2022
- [3] SoloParkour: Constrained Reinforcement Learning for Visual Locomotion from Privileged Experience, CoRL2024
- [4] Bootstrapping reinforcement learning with imitation for vision-based agile flight, CoRL2024
- [5] Privileged Sensing Scaffolds Reinforcement Learning. ICLR2024
- [6] Real-World Humanoid Locomotion with Reinforcement Learning, 2023
- [7] Bridging the Imitation Gap by Adaptive Insubordination, NeurIPS2021

**Questions:**

- Why does the "w/o align" baseline perform so well compared to BC and DAgger, particularly in the quadrotor environment? It seems this baseline would produce behaviors similar to that of BC and DAgger.
- Does the proposed method require more interactions than the baselines? Were baselines evaluated with equivalent budgets of privileged and high-dimensional samples?
- What is the performance without using a shared action decoder?
- How sensitive is the method to the weight of the KL divergence penalty term?
- Is it possible to have videos of the experiments in simulation ?
- Out of curiosity, have you experimented with using forward vs reverse KL divergence as the penalty term?

---

> ### Author Response · Authors · 2024-11-25
> **Official Comment by Authors (1/2)**
>
> **W1: Missing important comparison and discussion: Several prior works focus on training the student with RL to achieve behaviors distinct from the teacher. While some of these are referenced in the paper, there is no thorough discussion of this class of methods, and the authors only compare their approach to BC and DAgger, which were not specifically designed to handle observation asymmetry.**
>
> We sincerely appreciate the list of key related works, which has helped us better contextualize our contributions. In our revised manuscript, we have included a detailed discussion to position our work in relation to them. Specifically, most of the referenced works [1-7] assume a fixed teacher. This is in contrast to our proposed method, which adapts the teacher to the capabilities of the student.
>
> We have included a comparison with “Deep Whole-Body Control” [C1], which adjusts the teacher encoder by enforcing a shared feature space. Additionally, we also compare against the proposed “Real-World Humanoid Locomotion with Reinforcement Learning” [C2] method (currently only for vision-based manipulation). The results clearly show that our proposed method outperforms them. Additionally, we have expanded the experimental section to provide further insights into the results and the observed improvements.
>
> | **Methods**        | **Success Rate (Manipulation)** | **Success Rate (Quadrotor)** |
> |---------------------|----------------------------|----------------------------|
> | BC                 | 0.16 ± 0.15               | 0.05 ± 0.04               |
> | DAgger             | 0.34 ± 0.31               | 0.08 ± 0.03               |
> | **HLRL**               | 0.61 ± 0.22               | tbd                          |
> | **DWBC**               | 0.63 ± 0.18               | 0.35 ± 0.07               |
> | w/o Align (Ours)   | 0.61 ± 0.18               | 0.38 ± 0.11               |
> | w Align (Ours)     | **0.88 ± 0.07**           | **0.46 ± 0.04**           |
> #### **Updated baseline experiments.**
>
>
> [C1] Deep Whole-Body Control: Learning a Unified Policy for Manipulation and Locomotion, CoRL 2022
>
> [C2] Real-World Humanoid Locomotion with Reinforcement Learning, Science Robotics 2024
>
> ---
>
> **W2: Clarity and methodological details: Certain aspects of the approach, especially in Section 4, lack clarity...A potential drawback of this approach is that alternating training for both teacher and student before teacher training is complete might demand more high-dimensional samples than BC or DAgger**
>
> We enhanced the clarity of the methodology and experiments by restructuring sections, improving descriptions, and providing additional technical details.
>
> Indeed, training a teacher in our framework requires more training samples since the teacher policy must adapt to the evolving capabilities of the student while still accomplishing the task. However, the computational and time bottleneck lies in the student training, particularly in rendering high-dimensional images. Therefore, the additional (cheap) teacher interactions using low-dimensional observations can help to reduce the actual bottleneck of training a high-dimensional student.

---

> ### Author Response · Authors · 2024-11-25
> **Official Comment by Authors (2/2)**
>
> **Q1: Why does the "w/o align" baseline perform so well compared to BC and DAgger, particularly in the quadrotor environment? It seems this baseline would produce behaviors similar to that of BC and DAgger.**
>
> The main difference between our method without alignment and the imitation learning baselines is that we used a shared action decoder between the teacher and the student policies. Instead of aligning only the actions, we also align the embeddings of the action decoders. Similar to the results shown in [1][2], we believe that aligning both the actions and the observation representations will help the policy perform better than aligning only the actions.
>
> We would also like to emphasize that for the task of quadrotor obstacle avoidance, due to the evaluation period being rather short, the success rate information does not sufficiently capture the policy behavior. Hence, we have now introduced two new evaluation metrics describing how "environment aware" the resulting policies are. For example, the vision-based student policy will need to always look in the direction it flies towards to ensure it does not collide. This behavior is not learnable without changing the teacher policy's behavior. We observed that our approach still outperforms the baseline approaches, and the baseline without alignment performs similarly to the DAgger and BC baselines.
>
> | **Methods**        | **Velocity Angle [°] ↓** | **Num. Obstacle in View ↑** |
> |---------------------|--------------------------|-----------------------------|
> | BC                 | 75.5                    | 1.92                        |
> | DAgger             | 78.6                    | 2.42                        |
> | DWBC               | 46.7                    | 2.33                        |
> | w/o Align (Ours)   | 63.2                    | 2.61                        |
> | w Align (Ours)     | **32.2**                | **3.51**                    |
> #### **Perception-Aware Experiments.**
>
> [1] RMA: Rapid Motor Adaptation for Legged Robots, RSS 2021
>
> [2] In-Hand Object Rotation via Rapid Motor Adaptation, CoRL 2022
>
> ---
>
> **Q2: Does the proposed method require more interactions than the baselines? Were baselines evaluated with equivalent budgets of privileged and high-dimensional samples?**
>
> We are grateful for raising this concern. All the tested methods are evaluated with the exact number of privileged and high-dimensional samples. The different teachers for the imitation learning method are taken from the w/o alignment experiments, which train the teacher purely based on the RL reward. The baselines are then trained with the same number of rendered images as used in our approach.
> We have added this clarification to the revised manuscript.
>
> **Q3: What is the performance without using a shared action decoder?**
>
> We performed ablation experiments in the vision-based manipulation task. Without the shared action decoder, the performance of the vision-based student drops from 0.88 to 0.62 success rate. This confirms that the low-level task information can be shared between privileged teacher and student. Thus, the learned student policy benefits by also leveraging the teacher experiences for the shared action decoder.
>
> | **Configuration**   | **Success Rate**       |
> |-------------------------------|-------------------------|
> | w Reward / wo Loss       | 0.47 ± 0.37           |
> | wo Reward / w Loss       | 0.74 ± 0.08           |
> | wo shared decoder        | 0.62 ± 0.27           |
> | λ = 0.1                  | 0.81 ± 0.20           |
> | λ = 0.075                | 0.77 ± 0.18           |
> | (Default) λ = 0.05       | 0.88 ± 0.07           |
> | λ = 0.025                | 0.95 ± 0.03           |0.95 ± 0.03   |
> #### **Ablation Experiments.**
>
> **Q4: How sensitive is the method to the weight of the KL divergence penalty term?**
>
> Thank you for suggesting this ablation. We conducted experiments for the vision-based manipulation task to evaluate the impact of different weightings for the KL divergence term (0.1, 0.075, 0.05, 0.025). The results show that the vision-based student consistently outperforms the baseline across all tested values, with the highest performance achieved at a weighting factor of 0.025. Overall, our method demonstrated robustness to this hyperparameter, consistently achieving higher success rates than the baseline.
>
> ---
>
> **Q5: Is it possible to have videos of the experiments in simulation ?**
>
> We have included a video showcasing the performance for the two complex tasks: vision-based quadrotor flight and vision-based manipulation.
>
> ---
>
> **Q6: Out of curiosity, have you experimented with using forward vs reverse KL divergence as the penalty term?**
>
> That is an interesting question. We have not yet conducted such experiments due to time constraints. But we can include the result of such an experiment in the case of acceptance for the final version.

---

> ### Author Response · Authors · 2024-11-29
> **A gentle reminder**
>
> Thanks again for serving as a reviewer, we really appreciate your comments. Your feedback has strongly improved our paper, and we believe that we have addressed all of your concerns as we now have included 3 more baselines, 3 more ablations and substantially improved the writing. The end of the discussion period is rapidly approaching, and we would really appreciate it if you could check our response and let us know whether your concerns are well addressed. If not or in case you have any further concerns we would be more than happy to work with you on improving the paper.

---

> ### Comment · Reviewer_Xbsv · 2024-12-02
>
> Thank you for your response, the additional experiments, and the accompanying videos.
>
> The paper has seen improvements, and my primary concern has been reasonably addressed: the method is now better validated and contextualized.
> I am inclined to raise my score, acknowledging the authors' efforts.
>
> That said, there are still weaknesses:
> - While the clarity has improved, the paper remains somewhat hard to follow, despite the underlying method being conceptually simple in my opinion.
> - Watching the videos, the experimental results are a bit underwhelming. The observation asymmetry appears artificial: it is created in the manipulation environment via a weird camera angle, and in the quadrotor environment, it could arguably be addressed by adding a heading-related reward term.
> Introducing a more practical robotic environment, where achieving perception-aware behaviors presents a genuine challenge, would make the paper truly compelling.
>
> I encourage the authors to continue improving their paper, as it is still borderline in my opinion.
> Below, I have included a few suggestions to improve the writing, mostly centered on the method sections (these are subjective, and I defer to the authors' judgment in considering them).
>
> ---
>
> - From my point of view, the main contributions of the paper are the KL-regularization of the teacher and the introduction of a proxy student. These should be prominently highlighted and explained clearly. The remaining elements, while relevant, are secondary and could be streamlined. Simplifying and condensing the explanation of the method overall would be beneficial.
> - KL-regularized RL is a well-established technique, ex: [8-16] and many more. Including a discussion would be appropriate.
> - The derivations in Eq. 3–8 are straightforward and might not be necessary in the main paper. Summarizing them into a single equation and moving the detailed steps to the appendix would be more concise. Additionally, these derivations seem familiar and may already exist in the literature, potentially in one of [8-16].
> - Including more references is better in general, and I feel the initial submission had relatively few for this type of paper.
> - Figure 1 is difficult to understand.
> - The backpropagation graph is still unclear.
> - In Section 4, adding more equations that explicitly detail the training losses for each network would be helpful.
> - In particular, the integration and training of the shared action network is still unclear.
> - Algorithm 1 is quite helpful. It could be refined and moved to the main paper.
> - The explanations and the derivation of the KL divergence between two Gaussians (Eq. 9,10) are trivial and would be better suited to the appendix.
> - Having three separate "method" sections (Sec. 2,3,4) looks slightly unusual to me. Combining these into one or two sections might improve readability, flow and conciseness.
> - Please also add implementation details for the baselines, for example in the appendix.
>
> References:
> - [8] Optimal control as a graphical model inference problem, 2012
> - [9] Reinforcement learning and control as probabilistic inference: Tutorial and review, 2018
> - [10] Information asymmetry in kl-regularized RL, 2019
> - [11] Exploiting hierarchy for learning and transfer in kl-regularized rl, 2019
> - [12] Leverage the Average: an Analysis of KL Regularization in Reinforcement Learning, 2020
> - [13] Accelerating online reinforcement learning with offline datasets, 2020
> - [14] Accelerating reinforcement learning with learned skill priors, 2020
> - [15] Training language models to follow instructions with human feedback, 2022
> - [16] Soft Actor-Critic: Off-Policy Maximum Entropy Deep Reinforcement Learning with a Stochastic Actor, 2018

---

### Official Review · Reviewer_mh2n · 2024-11-03

**Soundness:** 2
**Presentation:** 2
**Contribution:** 2
**Rating:** 6
**Confidence:** 4

**Summary:**

The paper proposes to tackle the problem of information asymmetry in the privileged training and distillation paradigm by proposing to adapt the teacher policy’s behavior to account for the student’s partial observability by adding a reward term to encourage imitability of actions. They additionally introduce a proxy student network that approximates how the student behaves conditioned on the teacher’s privileged observations to alleviate the need of generating potentially high-dimensional student observations for optimizing the teacher policy.

**Strengths:**

The motivation of the problem and proposed approach is presented intuitively (but lacks some clarity see clarification questions Q1-2).

**Weaknesses:**

* The paper’s positioning in the related work as the only one considering information asymmetry in student-teacher framework is incorrect. There are several recent works that have looked at this problem [1-5], in fact the imitability cost like the one proposed is explored in [3].

* As a consequence of the above, I think the paper misses key baselines that actually tackle problems under similar assumptions of asymmetry (the considered BC and DAgger baselines are bound to fail here) and therefore does not establish novelty.

* Additionally, the introduced approach involves several hyperparameters – the balancing term for “imitability” cost, the update style (frequency/batch sizes) of the alignment phase. The paper does not describe the implications of these choices of the algorithm and in-reality these can be considerably hard to tune for each task and can raise concerns of applicability and reproducibility of the approach.

_References_:

[1] Warrington, Andrew, et al. "Robust asymmetric learning in POMDPs." International Conference on Machine Learning. PMLR, 2021.

[2] Weihs, Luca, et al. "Bridging the imitation gap by adaptive insubordination." Advances in Neural Information Processing Systems 34 (2021): 19134-19146.

[3] Nguyen, Hai, et al. "Leveraging fully observable policies for learning under partial observability." arXiv preprint arXiv:2211.01991 (2022).

[4] Walsman, Aaron, et al. "Impossibly good experts and how to follow them." The Eleventh International Conference on Learning Representations. 2022.

[5] Shenfeld, Idan, et al. "TGRL: An algorithm for teacher guided reinforcement learning." International Conference on Machine Learning. PMLR, 2023.

**Questions:**

(Q1) The description of the alignment phase (Sec 4.2) can benefit from more clarity: What subset of experiences are good enough to train the proxy student – how much divergence between student/student-proxy is tolerable? If this has to be a large subset of the data then there is no benefit from using a proxy student, one might as well use all the observations to update both the teacher and student. I think clarity of the approach will improve with a pseudocode description of all the phases with input requirements in the appendix.

(Q2) In the experiments, the baselines aren’t described clearly – what does w/o alignment mean? Does it suggest removal of proxy student network in (a) no imitability loss for teacher (so standard distillation setup) (b) imitability loss with actual student network. If (a), then why is the method “w/o align (ours)” in the result tables? Also, can the authors explain why the teacher returns w/o alignment are lower than with alignment in Figure 4, intuition suggests that it should be higher?

---

> ### Author Response · Authors · 2024-11-25
> **Official Comment by Authors (1/2)**
>
> **W1: The paper’s positioning in the related work as the only one considering information asymmetry in student-teacher framework is incorrect. There are several recent works that have looked at this problem [1-5], in fact the imitability cost like the one proposed is explored in [3].**
>
> We sincerely appreciate the list of key related works, which has helped us better contextualize our contributions. In our revised manuscript, we have included a detailed discussion to position our work in relation to them. Specifically, most of the referenced works [2, 3, 4, 5] assume a fixed teacher, with the exception of [1], which proposes an adaptive DAgger inside a POMDP formulation using a belief state. Similarly to our formulation, [3] employs the KL divergence between the teacher and student actions as a divergence metric. However, in [3], this divergence metric is used exclusively to train the student by a weighted combination of imitating expert actions and optimizing for RL rewards.
>
> ---
>
> **W2: As a consequence of the above, I think the paper misses key baselines that actually tackle problems under similar assumptions of asymmetry**
>
> Thank you for raising this point. Based on comments from the other reviewers, we have now included additional baselines that follow similar assumptions. We have included a comparison with “Deep Whole-Body Control” [1], which adjusts the teacher encoder by enforcing a shared feature space. Additionally, we also compare against the proposed “Real-World Humanoid Locomotion with Reinforcement Learning” method [2] (currently only for vision-based manipulation). The results clearly show that our proposed method outperforms them.
>
> | **Methods**        | **Success Rate (Manipulation)** | **Success Rate (Quadrotor)** |
> |---------------------|----------------------------|----------------------------|
> | BC                 | 0.16 ± 0.15               | 0.05 ± 0.04               |
> | DAgger             | 0.34 ± 0.31               | 0.08 ± 0.03               |
> | **HLRL**               | 0.61 ± 0.22               | tbd                          |
> | **DWBC**               | 0.63 ± 0.18               | 0.35 ± 0.07               |
> | w/o Align (Ours)   | 0.61 ± 0.18               | 0.38 ± 0.11               |
> | w Align (Ours)     | **0.88 ± 0.07**           | **0.46 ± 0.04**           |
> #### **Updated baseline experiments.**
>
> [1] Deep Whole-Body Control: Learning a Unified Policy for Manipulation and Locomotion, CoRL 2022
>
> [2] Real-World Humanoid Locomotion with Reinforcement Learning, Science Robotics 2024
>
> ---
>
> **W3: Additionally, the introduced approach involves several hyperparameters – the balancing term for “imitability” cost, the update style (frequency/batch sizes) of the alignment phase.**
>
> We conducted ablation experiments for the vision-based manipulation task to evaluate the impact of different weightings for the KL divergence term (0.1, 0.075, 0.05, 0.025). The results show that the vision-based student consistently outperforms the baseline across all tested values, with the highest performance achieved at a weighting factor of 0.025. Overall, our method demonstrated robustness to this hyperparameter, consistently achieving higher success rates than the baseline.
>
> | **Configuration**   | **Success Rate**       |
> |-------------------------------|-------------------------|
> | w Reward / wo Loss       | 0.47 ± 0.37           |
> | wo Reward / w Loss       | 0.74 ± 0.08           |
> | wo shared decoder        | 0.62 ± 0.27           |
> | λ = 0.1                  | 0.81 ± 0.20           |
> | λ = 0.075                | 0.77 ± 0.18           |
> | (Default) λ = 0.05       | 0.88 ± 0.07           |
> | λ = 0.025                | 0.95 ± 0.03           |0.95 ± 0.03   |
> #### **Ablation Experiments.**

---

> ### Author Response · Authors · 2024-11-25
> **Official Comment by Authors (2/2)**
>
> **Q1: The description of the alignment phase (Sec 4.2) can benefit from more clarity: What subset of experiences are good enough to train the proxy student – how much divergence between student/student-proxy is tolerable? If this has to be a large subset of the data then there is no benefit from using a proxy student, one might as well use all the observations to update both the teacher and student. I think clarity of the approach will improve with a pseudocode description of all the phases with input requirements in the appendix.**
>
> We observed that our approach is generally robust to divergences between the student and proxy student. However, it is challenging to deterministically control or quantify this divergence, which limits our ability to provide a detailed quantitative analysis.
> Regarding sample efficiency, the subset of interactions required for the student is significantly smaller than the total interactions collected by the teacher. For instance, in the vision-based manipulation task, the teacher utilizes 98,304,000 interactions, whereas the student only requires 3,072,000 interactions. This represents a substantial 32-fold reduction in sample requirements.
> We are thankful for the recommendation to add a pseudocode description to the appendix, which we have included in Section A.2 in the appendix.
>
> ---
>
> **Q2: In the experiments, the baselines aren’t described clearly – what does w/o alignment mean? Does it suggest removal of proxy student network in (a) no imitability loss for teacher (so standard distillation setup) (b) imitability loss with actual student network. If (a), then why is the method “w/o align (ours)” in the result tables? Also, can the authors explain why the teacher returns w/o alignment are lower than with alignment in Figure 4, intuition suggests that it should be higher?**
>
> We appreciate the helpful feedback and have addressed these concerns by adding a dedicated subsection to introduce the baselines. In the “w/o alignment” setting, we exclude the KL-Divergence from both the reward and the policy update phase while keeping the paired L1-Loss on shared action decoder features. As a result, the student does not affect the teacher, and the proxy student is not needed. This configuration represents a standard distillation setup, leveraging identical network alignments to specifically evaluate the contributions of our proposed KL-Divergence terms in the reward and policy update.
>
> One plausible explanation for the increased teacher return observed in Figure 4 is that perception-aware behavior enhances the robustness of the teacher. With our framework, the teacher adopts safer behaviors, such as maintaining a greater distance from obstacles to account for the student's limitations. This can be observed in Figure 3 by the small distance of the teacher “without alignment” and the last obstacle. By reducing risk, the teacher achieves more consistent long-term returns.

---

> ### Author Response · Authors · 2024-11-29
> **A gentle reminder**
>
> Thanks again for serving as a reviewer, we really appreciate your comments. Your feedback has strongly improved our paper, and we believe that we have addressed all of your concerns as we now have included 3 more baselines, 3 more ablations and substantially improved the writing. The end of the discussion period is rapidly approaching, and we would really appreciate it if you could check our response and let us know whether your concerns are well addressed. If not or in case you have any further concerns we would be more than happy to work with you on improving the paper.

---

> ### Author Response · Authors · 2024-12-02
> **The discussion period ends today**
>
> Dear reviewer mh2n,
>
> With the end of the discussion period approaching (today), we have not yet received a response from you regarding our rebuttal. We would really appreciate it if you could review our response and revised manuscript at your earliest convenience. If our rebuttal has adequately addressed your concerns, we would greatly appreciate it if you could consider adjusting your score accordingly. Thank you once again for your time and for providing such valuable feedback.

---

> > ### Comment · Reviewer_mh2n · 2024-12-02
> >
> > Thank you for the detailed response. I appreciate the changes in the related work section to appropriately contextualize the paper's contribution and clarifications to my concerns. I am now in agreement with the novelty of the contribution and have raised my score in response to the new experiments comparing the proposed approach to more closely related works in the area. However, I feel that the paper in its current state is a little rushed and would encourage the authors to consider improving the presentation – especially providing more implementation details for the baselines.

---

### Official Review · Reviewer_xapm · 2024-11-04

**Soundness:** 3
**Presentation:** 3
**Contribution:** 3
**Rating:** 5
**Confidence:** 4

**Summary:**

The paper addresses the important problem of information asymmetry between teacher and student policies in imitation learning. The authors propose a novel joint training framework that encourages the teacher to learn behaviors that can be more easily imitated by the student, thereby mitigating the negative impact of the information gap. The core idea is to incorporate the performance bound of imitation learning into the teacher's objective function, resulting in two key modifications: a KL-divergence-based penalty term in the teacher's reward function and a KL-divergence-based supervisory signal for updating the teacher's network parameters. The effectiveness of the proposed method is validated through experiments on three diverse tasks: maze navigation, vision-based quadrotor obstacle avoidance, and vision-based drawer opening with a robotic arm.

**Strengths:**

1. The paper presents a novel approach to tackling the teacher-student information asymmetry problem by incorporating the imitation learning performance bound into the teacher's objective function, leading to a creative combination of ideas from imitation learning theory and practical algorithm design.
2. The proposed method is well-motivated and grounded in the theoretical foundations of imitation learning, with clear explanations for the design of the KL-divergence-based penalty and supervisory terms.
3. The method is evaluated on three diverse tasks, demonstrating its applicability to both discrete and continuous control domains. The results show improvements over baseline imitation learning approaches.
4. The paper is generally well-written and easy to follow, with a clear problem statement, detailed method description, and effective use of figures and tables.

**Weaknesses:**

1. Limited theoretical analysis: The paper lacks a comprehensive theoretical analysis comparing the proposed method to existing approaches. A more rigorous theoretical justification for the superiority of the method would strengthen the contributions.
2. Insufficient ablation studies: The individual contributions of the reward penalty and the KL-divergence supervision are not clearly distinguished through ablation experiments, making it difficult to assess the necessity of each component.
3. Limited experimental evaluation: While the method is evaluated on three tasks, a more extensive experimental evaluation on a wider range of environments and benchmarks would provide stronger evidence for the generalizability and effectiveness of the approach.
4. Clarity of certain technical details: Some aspects of the paper, such as the definition of the proxy student network and the justification for the equality of covariance matrices in equations (9) and (10), require further clarification.

**Questions:**

1. How does the proposed method theoretically compare to existing approaches in terms of convergence guarantees and sample complexity?
2. What is the sensitivity of the method to the choice of the KL-divergence penalty weight, and how can this hyperparameter be effectively tuned?
3. Can the authors provide more detailed ablation studies to demonstrate the individual contributions of the reward penalty and the KL-divergence supervision?
4. How well does the method scale to more complex environments and tasks, and what are the potential limitations in terms of computational overhead and sample efficiency?
5. Can the authors discuss the potential extensions of the proposed framework to other imitation learning algorithms and settings, such as inverse reinforcement learning or multi-agent imitation learning?

---

> ### Author Response · Authors · 2024-11-25
> **Official Comment by Authors (1/2)**
>
> **W1: Limited theoretical analysis: The paper lacks a comprehensive theoretical analysis comparing the proposed method to existing approaches.**
>
> We do agree that such a theoretical analysis would provide additional justification for our approach. Unfortunately, this analysis is beyond the scope of this work. Nevertheless, we believe our experiments on two complex robotics tasks demonstrate the generality of the framework and its ability to improve student performance in challenging Imitation Learning settings. We leave a deeper theoretical investigation as an avenue for future work.
>
> ---
>
> **W2: Insufficient ablation studies: The individual contributions of the reward penalty and the KL-divergence supervision are not clearly distinguished through ablation experiments, making it difficult to assess the necessity of each component.**
>
> Based on the helpful feedback, we conducted additional ablation studies on the manipulation task. These experiments evaluate the contributions of both proposed components. The results show that each component independently enhances performance while also demonstrating low sensitivity to parameter variations.
>
> | **Configuration**   | **Success Rate**       |
> |-------------------------------|-------------------------|
> | w Reward / wo Loss       | 0.47 ± 0.37           |
> | wo Reward / w Loss       | 0.74 ± 0.08           |
> | wo shared decoder        | 0.62 ± 0.27           |
> | λ = 0.1                  | 0.81 ± 0.20           |
> | λ = 0.075                | 0.77 ± 0.18           |
> | (Default) λ = 0.05       | 0.88 ± 0.07           |
> | λ = 0.025                | 0.95 ± 0.03           |0.95 ± 0.03   |
> #### **Ablation Experiments.**
>
> ---
>
> **W3: Limited experimental evaluation**
>
> We have included a comparison with “Deep Whole-Body Control” [1] and “Real-World Humanoid Locomotion with Reinforcement Learning” [2] (currently only for vision-based manipulation) in the revised manuscript. The results clearly show that our proposed method outperforms both baselines. Additionally, we have expanded the experimental section to provide further insights into the results and the observed improvements.
>
> | **Methods**        | **Success Rate (Manipulation)** | **Success Rate (Quadrotor)** |
> |---------------------|----------------------------|----------------------------|
> | BC                 | 0.16 ± 0.15               | 0.05 ± 0.04               |
> | DAgger             | 0.34 ± 0.31               | 0.08 ± 0.03               |
> | **HLRL**               | 0.61 ± 0.22               | tbd                          |
> | **DWBC**               | 0.63 ± 0.18               | 0.35 ± 0.07               |
> | w/o Align (Ours)   | 0.61 ± 0.18               | 0.38 ± 0.11               |
> | w Align (Ours)     | **0.88 ± 0.07**           | **0.46 ± 0.04**           |
> #### **Updated baseline experiments.**
>
> [1] Deep Whole-Body Control: Learning a Unified Policy for Manipulation and Locomotion, CoRL 2022
>
> [2] Real-World Humanoid Locomotion with Reinforcement Learning, Science Robotics 2024
>
> ---
>
> **W4: Clarity of certain technical details: Some aspects of the paper, such as the definition of the proxy student network and the justification for the equality of covariance matrices in equations (9) and (10), require further clarification**
>
> We clarified the definition of the proxy student as well as the equality of the covariance matrices in equations (9) and (10).

---

> ### Author Response · Authors · 2024-11-25
> **Official Comment by Authors (2/2)**
>
> **Q1: How does the proposed method theoretically compare to existing approaches in terms of convergence guarantees and sample complexity?**
>
> We agree that a stronger theoretical analysis of our method might be beneficial, but we believe it would be beyond the scope of this work.  Nevertheless, our experiments on two complex robotics tasks demonstrate the generality of the framework and its ability to improve student performance in challenging Imitation Learning settings. We leave a deeper theoretical investigation as an avenue for future work.
>
> ---
>
> **Q2: What is the sensitivity of the method to the choice of the KL-divergence penalty weight, and how can this hyperparameter be effectively tuned?**
>
> Thank you for suggesting this interesting ablation. We conducted experiments for the vision-based manipulation task to evaluate the impact of different weightings for the KL divergence term (0.1, 0.075, 0.05, 0.025). The results show that the vision-based student consistently outperforms the baseline across all tested values, with the highest performance achieved at a weighting factor of 0.025. Overall, our method demonstrated robustness to this hyperparameter, consistently achieving higher success rates than the baseline.
>
> ---
>
> **Q3: Can the authors provide more detailed ablation studies to demonstrate the individual contributions of the reward penalty and the KL-divergence supervision?**
>
> Thank you for this valuable suggestion. We performed this ablation study on the vision-based student in the manipulation task. Namely, we exclude the reward penalty while keeping the KL divergence in the teacher updates, the success rate drops from 0.88 to 0.74. This performance drop is even larger (from 0.88 to 0.47) when the KL divergence in the teacher update is removed while keeping the reward penalty. These results suggest that feature alignment between the teacher and student, driven by the KL divergence, has a greater impact on the imitability of the teacher's behavior. The highest success rate is achieved when both the KL divergence and the reward penalty are used to shape the behavior of the teacher.
>
> ---
>
> **Q4: How well does the method scale to more complex environments and tasks, and what are the potential limitations in terms of computational overhead and sample efficiency?**
>
> We believe that the tasks of vision-based obstacle avoidance with a quadrotor and vision-based manipulation offer complex environments that are highly relevant to real-world applications. The manipulation task requires the policy to control a robot with complex kinematics and high degrees of freedom, and to manage contact interactions, whereas the quadrotor task involves a mobile robot performing agile movements to avoid obstacles, which necessitates handling highly non-linear dynamics.
> Our proposed method can readily be applied across both tasks without major modifications. Therefore, we do not anticipate any additional computational overhead.
>
> ---
>
> **Q5: Can the authors discuss the potential extensions of the proposed framework to other imitation learning algorithms and settings, such as inverse reinforcement learning or multi-agent imitation learning?**
>
> Our proposed framework focuses on adapting the teacher training to the capabilities of the student, which sets it apart from most existing imitation learning methods that primarily train the student without modifying the teacher. Consequently, our approach can be combined with many existing imitation learning algorithms to better align the student with the teacher. This is especially relevant for IL focusing more on representation learning.
>
> Extending our framework to multi-agent imitation learning is also feasible as long as access to the teacher policy is available during training. In such scenarios, both teachers and students can be trained simultaneously without requiring significant changes to the training pipeline.
>
> In contrast, applying our approach to the inverse reinforcement learning setting is not straightforward. Since IRL assumes the policy generating the demonstrations is unknown, our framework relying on adapting the teacher cannot be directly applied in such settings.

---

> ### Author Response · Authors · 2024-11-29
> **A gentle reminder**
>
> Thanks again for serving as a reviewer, we really appreciate your comments. Your feedback has strongly improved our paper, and we believe that we have addressed all of your concerns as we now have included 3 more baselines, 3 more ablations and substantially improved the writing. The end of the discussion period is rapidly approaching, and we would really appreciate it if you could check our response and let us know whether your concerns are well addressed. If not or in case you have any further concerns we would be more than happy to work with you on improving the paper.

---

> ### Author Response · Authors · 2024-12-02
> **The discussion period ends today**
>
> Dear reviewer xapm,
>
> With the end of the discussion period approaching (today), we have not yet received a response from you regarding our rebuttal. We would really appreciate it if you could review our response and revised manuscript at your earliest convenience. If our rebuttal has adequately addressed your concerns, we would greatly appreciate it if you could consider adjusting your score accordingly. Thank you once again for your time and for providing such valuable feedback.

---

### Official Review · Reviewer_2qRa · 2024-11-09

**Soundness:** 2
**Presentation:** 2
**Contribution:** 3
**Rating:** 6
**Confidence:** 5

**Summary:**

The authors propose to improve imitation learning between a privileged expert policy that sees more information and a partially observed student policy. They propose to train the teacher policy so that it maximizes return while staying close to the student policy distribution, so that the teacher and student trajectory distributions are more aligned. The authors show this improves performance in several simulated robotic setups.

---
Post rebuttal - the authors have done a decent job addressing my concerns on experimentation and improved the presentation.

Updated my score to 6.

I do agree with reviewer mh2n that this submission still feels a bit rushed, with a lot of important results (comparison to competitive baselines and ablations) coming in through the rebuttal period. I tend to be a bit more wary of results that come in through rebuttal, given the limited time and resources.

**Strengths:**

- Simple approach to handling asymmetry in teacher student policy distillation, where teacher is trained to minimize divergence between teacher and student. this will constrain the teacher to states where the student can explore
- Interesting experimental findings.
	- The experiments show that teacher policies trained in this manner are better for teaching student policies.
	- A teacher with this constraint gets higher returns than a teacher without, because learning to act with less inputs leads to more robustness and generalization behavior. This would be really interesting to explore further, with more experiments or analysis.

**Weaknesses:**

## Experimental section has several problems
- Experimental section is lacking, sparse in task selection and choices of baseline, and seem a bit contrived. Several ways to fix this:
	- More standardized benchmarks, using envs from prior work like  [1, 2]
	- better baselines and analysis (see below)
	-  Sim2real of the drone / manipulator results would round it out

- choice of baselines is lacking, really should compare with prior work in handling asymmetric RL / IL problems ( see references below)
- Baseline details are completely missing, even if BC and Dagger are widely known, their training details need to be written down.
- little-to-no ablations, i.e. why the shared feature encoder? Wouldn't we expect that the teacher representation be very different than the student representation in very partially observed settings?

- The authors need to put more work in the experimental section, which is verbose and unorganized. Having two giant paragraphs per experiment, one for setup, and one for results, is lazy and hard for readers to parse. Would like to see more figures, analysis of the teacher and student behavior, especially in the drone and manipulator case. IMO, the color maze experiment, which is a toy experiment, does not require that much text and space dedicated to explaining it. It could even be moved to appendix.
- Another way to improve the experimental section, is to provide some real world robot results. Because this is ICLR (more focused on learning methods), I would say this isn't a strict requirement, and I would appreciate more rigorous comparisons in the previous experiments. But real robot results would definitely round out the experimental section.

## Another weakness - having to train a teacher policy from scratch
- One drawback of this paper is the need to train a teacher policy with their specialized objective. This makes the method harder to use in practice since this method requires training a teacher policy from scratch using RL and many, many samples in a fast simulator  (10^8 for maze, 100M for manipulation). Getting a good simulator and doing sim2real is not always feasible.
	- In contrast, other methods like ADVISOR, COSIL, etc. (see references below) assume that the teacher policy is given, and do not require teacher training. In robotics,  where reasonable teacher policies can be obtained through scripting and pre-existing controllers,  it seems that approaches that do not modify the teacher are easier to use.

Some discussion here would be appreciated.


## Minor: missing connections to prior work on regularizing privileged teachers and representations
- Regularizing the privileged policy / representation so that it doesn't stray too far away from the student  has been explored in the past, see [6,7,8]

1. Leveraging Fully Observable Policies for Learning under Partial Observability
2. Privileged Sensing Scaffolds Reinforcement Learning
3. Bridging the Imitation Gap by Adaptive Insubordination
4. TGRL: An Algorithm for Teacher Guided Reinforcement Learning
5. Impossibly Good Experts and How to Follow Them
6. Bridging the Sim-to-Real Gap from the Information Bottleneck Perspective
7. Deep Whole-Body Control: Learning a Unified Policy for Manipulation and Locomotion
8. Designing Skill-Compatible AI: Methodologies and Frameworks in Chess

**Questions:**

See weaknesses above. I am eager to see the paper be improved.

---

> ### Author Response · Authors · 2024-11-25
> **Official Comment by Authors (1/2)**
>
> **W1: Experimental section is lacking, sparse in task selection and choices of baseline, and seem a bit contrived.**
>
> We have introduced additional baselines, conducted a more detailed analysis of the impact of each proposed component, and provided further insights in the experimental section.
>
> ---
>
> **W2: Choice of baselines is lacking.**
>
> We ran additional experiments with 2 new baselines, namely  “Deep Whole-Body Control” [5] and “Real-World Humanoid Locomotion with Reinforcement Learning” [6] (currently only for the vision-based manipulation task) . Our results demonstrate that our proposed method significantly outperforms these baselines.
>
> | **Methods**        | **Success Rate (Manipulation)** | **Success Rate (Quadrotor)** |
> |---------------------|----------------------------|----------------------------|
> | BC                 | 0.16 ± 0.15               | 0.05 ± 0.04               |
> | DAgger             | 0.34 ± 0.31               | 0.08 ± 0.03               |
> | **HLRL**               | 0.61 ± 0.22               | tbd                          |
> | **DWBC**               | 0.63 ± 0.18               | 0.35 ± 0.07               |
> | w/o Align (Ours)   | 0.61 ± 0.18               | 0.38 ± 0.11               |
> | w Align (Ours)     | **0.88 ± 0.07**           | **0.46 ± 0.04**           |
> #### **Updated baseline experiments.**
> ---
>
> **W3: Baseline details are completely missing.**
>
> Thank you for pointing that out. We added a new subsection to the experiments section to introduce the tested baselines, ensuring greater clarity and completeness.
>
> ---
>
> **W4: little-to-no ablations, i.e. why the shared feature encoder? Wouldn't we expect that the teacher representation be very different than the student representation in very partially observed settings.**
>
> We ran additional experiments to perform all requested ablations. The results show that the shared action decoder is a critical component, increasing the success rate from 0.62 to 0.88. We argue that the teacher encoder learns high-level task representations rather than perception-specific features. With our KL-Divergence loss in the teacher policy updates, we ensure that the teacher encoder extracts task-relevant features that can be learned by the student.
>
> | **Configuration**   | **Success Rate**       |
> |-------------------------------|-------------------------|
> | w Reward / wo Loss       | 0.47 ± 0.37           |
> | wo Reward / w Loss       | 0.74 ± 0.08           |
> | wo shared decoder        | 0.62 ± 0.27           |
> | λ = 0.1                  | 0.81 ± 0.20           |
> | λ = 0.075                | 0.77 ± 0.18           |
> | (Default) λ = 0.05       | 0.88 ± 0.07           |
> | λ = 0.025                | 0.95 ± 0.03           |0.95 ± 0.03   |
> #### **Ablation Experiments.**
>
> ---
>
> **W5: The authors need to put more work in the experimental section**
>
> Thank you for pointing that out. We restructured the experimental section to improve readability and organization. Implementation details have been moved to the appendix, while a dedicated subsection introduces the tested baselines. Additionally, the section now includes ablation studies.
>
> ---
>
> **W6: Another way to improve the experimental section, is to provide some real world robot results. Because this is ICLR (more focused on learning methods), I would say this isn't a strict requirement**
>
> Both tasks are highly complex robotics challenges that require sophisticated hardware and experimental setups for real-world testing. However, the simulation frameworks we employ have been shown to successfully transfer to real-world scenarios [1][2][3].
>
> [1] Champion-level Drone Racing using Deep Reinforcement Learning, Nature, 2023
>
> [2] Reaching the Limit in Autonomous Racing: Optimal Control vs. Reinforcement Learning, Science Robotics, 2023
>
> [3] On the role of the action space in robot manipulation learning and sim-to-real transfer, obotics and Automation Letters, 2024

---

> ### Author Response · Authors · 2024-11-25
> **Official Comment by Authors (2/2)**
>
> **W7: One drawback of this paper is the need to train a teacher policy with their specialized objective. This makes the method harder to use in practice since this method requires training a teacher policy from scratch using RL and many, many samples in a fast simulator (10^8 for maze, 100M for manipulation). Getting a good simulator and doing sim2real is not always feasible.**
>
> Indeed, training a teacher in our framework requires more training samples since the teacher policy must adapt to the evolving capabilities of the student while still accomplishing the task. However, the computational and time bottleneck lies in the student training, particularly in rendering high-dimensional images. Therefore, the cheap teacher interactions using low-dimensional observations can help to reduce the actual bottleneck of training a high-dimensional student.
> As correctly noted, our method depends on training the teacher with RL. However, privileged teacher training using RL  is a common practice nowadays in the domain of robotics [1][2][3][4], which leads to robust policy performance in both simulation and the real world.
> We have added the assumption of joint teacher and student training to the revised manuscript.
>
> [1] Learning high-speed flight in the wild, Science Robotics 2021
>
> [2] Learning robust perceptive locomotion for quadrupedal robots in the wild, Science Robotics 2022
>
> [3] Extreme Robot Parkour, ICRA 2024
>
> [4] Humanoid Parkour Learning, CoRL 2024
>
> [5] Deep Whole-Body Control: Learning a Unified Policy for Manipulation and Locomotion, CoRL2022
>
> [6] Real-World Humanoid Locomotion with Reinforcement Learning, Science Robotics 2024

---

> > ### Comment · Reviewer_2qRa · 2024-11-27
> > **Thanks for the update, still have remaining questions.**
> >
> > Hello,
> > I thank the authors for their responses, which will be helpful for my final decision. I still have some questions on differences between proposed method and actual implementation.
> >
> > It seems like the objective prescribes a KL divergence term to keep the teacher close to the student. But in section 4.2 Rollout phase, the teacher is trained on an additional penalty term based on the action difference between teacher and proxy student. Shouldn't the KL divergence term be enough? Why add the additional penalty term, and why is it helpful over just using KL?
> >
> > Next, there are some details on shared networks and updates to particular parts of the network that I'm not clear on. It seems like the student and teacher action decoders are shared. The action decoding layers are only updated on the policy gradient computed with the task reward. What is going on here? Are the encoders getting gradients from the full objective?
> >
> > It would be great if the authors can write out the exact computation graph somewhere, like what networks are getting which gradients, etc. The current architecture figure in the paper isn't too intuitive, especially the uni-directional arrows for the loss / KL don't clearly show me what is the prediction and what is the target.

---

> ### Author Response · Authors · 2024-11-27
> **Official Comment by Authors**
>
> We are glad that our responses have been helpful and appreciate the opportunity to provide further clarification.
>
> ---
>
> **It seems like the objective prescribes a KL divergence term to keep the teacher close to the student. But in section 4.2 Rollout phase, the teacher is trained on an additional penalty term based on the action difference between teacher and proxy student. Shouldn't the KL divergence term be enough? Why add the additional penalty term, and why is it helpful over just using KL?**
>
> The KL-term inside the reward term can be interpreted as a reward encouraging the teacher policy to visit states where the student and teacher are aligned and avoid states with a large misalignment. Thus, it affects the exploration of the teacher policy. In contrast, the KL-term in the objective/loss aligns the teacher action to the student action and can be interpreted as an alignment on the learned representation space. The positive effect of both terms is also confirmed by the ablation experiments in Table 2.
>
> ---
>
> **Next, there are some details on shared networks and updates to particular parts of the network that I'm not clear on. It seems like the student and teacher action decoders are shared. The action decoding layers are only updated on the policy gradient computed with the task reward. What is going on here? Are the encoders getting gradients from the full objective?**
>
> As correctly noted, the teacher, student, and proxy student share the same action decoder, which is updated only with the policy gradient computed from the task reward.  With the same policy gradient, the teacher encoder is updated. The student and proxy student are updated in the alignment phase. Importantly, during alignment, the teacher encoder remains fixed and is not updated. In the alignment phase, the following encoders are updated:
> - Student Encoder: Updated by aligning the student to the teacher while stopping the gradient flow to the teacher.
> - Proxy Student Encoder: Updated by aligning the proxy student to the image-based student, with the gradient flow stopped at the student.
>
> ---
>
> **It would be great if the authors can write out the exact computation graph somewhere, like what networks are getting which gradients, etc. The current architecture figure in the paper isn't too intuitive, especially the uni-directional arrows for the loss / KL don't clearly show me what is the prediction and what is the target.**
>
> Based on the feedback, we have introduced the gradient flow with different colors representing the different losses in Figure 1a). We have also introduced some clarification in the text to better highlight different updates. Additionally, we added a pseudocode algorithm in the appendix that illustrates better which network is updated in which phase. We remain open to further suggestions and sincerely thank the reviewer for their valuable input!

---

> > ### Comment · Reviewer_2qRa · 2024-11-27
> > **Clarifications continued**
> >
> > Thanks, this is helpful. Remaining questions:
> >
> > Task reward = just the environmental reward, or does it also include the KL penalty term?
> >
> > It's still not clear to me why you need to include both the KL penalty term in the rollout phase, and the KL divergence term in the loss. Aren't they somewhat redundant, i.e. if you take the policy gradient of the KL penalty term, it will give you a similar / same gradient as doing the KL divergence term between student and teacher policy distributions?
> >
> > My point of confusion is that the equation 8 just has one KL divergence term, which aims to make the student and teacher state marginals close. But now there are two KL terms in the implementation. Is one of the two a heuristic? If so, it would be nice to state clearly which term directly motivates the method, and which one is a heuristic.
> >
> > Overall, I think I get most of the algorithm now, but I would encourage the authors to continue working on the presentation of the algorithm, i.e. see if people outside the project can easily understand the algorithm from just reading the paper and figures.

---

> > > ### Author Response · Authors · 2024-11-28
> > > **Official Comment by Authors**
> > >
> > > We are grateful for the thoughtful clarification questions and discussion. In response, we have further refined the manuscript, placing greater emphasis on clearly distinguishing the different KL-Divergence terms to enhance understanding.
> > >
> > > ---
> > >
> > > **Task reward = just the environmental reward, or does it also include the KL penalty term?**
> > >
> > > That is correct, the task reward is just the environmental reward defined by the design of the task. It does not include the KL penalty term.
> > >
> > > ---
> > >
> > > **It's still not clear to me why you need to include both the KL penalty term in the rollout phase, and the KL divergence term in the loss. Aren't they somewhat redundant, i.e. if you take the policy gradient of the KL penalty term, it will give you a similar / same gradient as doing the KL divergence term between student and teacher policy distributions?**
> > >
> > > The two terms do seem similar at first glance, but they serve different purposes, namely, representation learning, and exploration. In theory, if you take the policy gradient of the KL penalty term, you would obtain the following term:
> > >
> > > $\int \nabla_\theta  \log p_{\theta}(\tau) [\sum_{s_t\in \tau}\gamma^t D_{KL}(\pi_T(\cdot|s_t),\pi_S(\cdot|s_t))].$
> > >
> > > While the KL divergence term in the loss is the expectation of the gradient of the KL divergence term:
> > >
> > > $\int p_{\theta}(\tau) \nabla_\theta  D_{\theta}(\tau) d\tau$
> > >
> > > These two terms are quite different. Also, in practice, adding the KL-term in the reward function provides a gradient signal that includes information about the long-term effect of taking certain actions on the divergence between student and teacher in future states. Hence the KL-divergence in the reward serves to modify the exploration behavior of the policy in a way that discourages discrepancy between student and teacher. In contrast, the KL-Divergence in the loss only measures the difference between student and teacher actions for the samples in the current batch (without explicitly looking at the effect on future states). Hence, this term only serves for learning representations that are similar between student and teacher and does not affect the exploration.
> > >
> > > ---
> > >
> > > **My point of confusion is that the equation 8 just has one KL divergence term, which aims to make the student and teacher state marginals close. But now there are two KL terms in the implementation. Is one of the two a heuristic? If so, it would be nice to state clearly which term directly motivates the method, and which one is a heuristic.**
> > >
> > > The two terms arise from the theoretical objective of minimizing the upper bound of the difference between teacher and student performance. Specifically, they emerge because of the product rule while taking the derivative with respect to the teacher weights, see Eq. 7. This leads to two KL divergence terms in Eq. 8, which are represented with D_theta in the Policy Gradient and KL-Div gradient. This theoretical derivation is further supported by the increased performance achieved while using both terms in the algorithm formulation, as reported in Table 2.

---

> > > > ### Comment · Reviewer_2qRa · 2024-11-29
> > > > **Followup**
> > > >
> > > > Thank you for handling the KL divergence questions, I understand it fully now. I am still unclear on what gradients the shared action decoder is getting, versus the encoders.
> > > >
> > > > Line 211: action decoder is just getting the task reward (i.e. no KL penalty?)
> > > >
> > > > Line 788 and 794: these lines suggest you're computing policy gradient wrt task reward + penalty term. And is the encoder getting gradients from task reward + penalty term + KL term, and the action decoder getting gradients from just the task reward? Why do we need to be so selective with the gradients, is it to prevent collapse due to the shared action decoder?
> > > >
> > > > Please clarify for me, what the action decoder is getting trained on, and what the encoders are getting trained on, with respect to all the terms in the objective.

---

> > > > > ### Author Response · Authors · 2024-11-30
> > > > > **Official Comment by Authors**
> > > > >
> > > > > We are happy to hear that we could clarify the KL divergence question and improve the manuscript based on the helpful discussions. We are also thankful for the continued help to improve the clarity of the manuscript, especially spotting an error in the text. We will correct and clarify this point in the next version of the manuscript.
> > > > >
> > > > > ---
> > > > >
> > > > > **Line 211: action decoder is just getting the task reward (i.e. no KL penalty?)**
> > > > >
> > > > > We are grateful for pointing out this mistake in the sentence in Line 211. The shared action decoder is trained with the policy gradient (task reward + KL penalty) and the KL-Div gradient (computed based on the KL divergence between proxy and teacher actions). Thus, it is trained with the same gradients as the teacher encoder. We will correct this point in the next version of the manuscript.
> > > > >
> > > > > ---
> > > > >
> > > > > **Line 788 and 794: these lines suggest you're computing policy gradient wrt task reward + penalty term. And is the encoder getting gradients from task reward + penalty term + KL term, and the action decoder getting gradients from just the task reward? Why do we need to be so selective with the gradients, is it to prevent collapse due to the shared action decoder?**
> > > > >
> > > > > That is correct; the policy gradient is computed based on the task reward and the penalty term, see also Eq. 8. The teacher encoder is trained with the same gradients as the shared task decoder, i.e., with the policy gradient (task reward + KL penalty) and the KL-Div gradient. As correctly stated, during the alignment phase, the shared task decoder is frozen to avoid the collapse of the feature space between the different encoders (teacher, student, proxy student). The collapse can happen since one perfect alignment between the three encoders is achieved by predicting a constant output, which harms the task performance.
> > > > >
> > > > > The different networks are trained based on the following gradients:
> > > > > - **Shared action decoder**: policy gradient (task reward + KL penalty), KL-Divergence gradient.
> > > > > - **Teacher encoder**: policy gradient (task reward + KL penalty), KL-Divergence gradient.
> > > > > - **Student encoder**: L1-Loss between student and frozen teacher network activations
> > > > > - **Proxy student encoder**: L1-Loss between proxy student and frozen student network activations.

---

### Author Response · Authors · 2024-11-25
**General Response**

We sincerely thank the reviewers for their valuable feedback regarding clarity, missing related work, and requests for additional ablation and baseline experiments. In response, we have made significant improvements to the manuscript (highlighted in blue), as summarized below:
- **Ablations**: We conducted additional experiments for the vision-based manipulation task to evaluate the impact of the proposed components. These results are now included in the updated manuscript.
- **Related Work**: We have incorporated all the missing references suggested by the reviewers and expanded the discussion to better contextualize our contributions.
- **Baselines**: To more effectively highlight the advantages and performance improvements of our method, we have added results for baseline methods. Given the complexity of the tasks, these comparisons further demonstrate the effectiveness of our approach.
- **Technical Clarity**: We enhanced the clarity of the methodology and experiments by restructuring sections, improving descriptions, and providing additional technical details.
- **Improved Stability**: We observed simulation instabilities for the manipulation task
when the Franka arm touches the top of the drawer (particularly with our proposed method). To address this, we have slightly lowered the position of the Franka robot, leading to more consistent behavior, as reported in the updated Table 2.

We look forward to further engaging discussions and appreciate the opportunity to refine our work based on the reviewers' constructive feedback.

---

> ### Author Response · Authors · 2024-11-28
> **Updated PDF Response**
>
> Based on the helpful feedback and discussions, we have extended the last submission to improve further the clarity and provide more details. In addition to the already reported ablation experiments, we have now included in total three baseline approaches for the vision-based manipulation task and two baseline approaches for the vision-based quadrotor task. We will add the third baseline to the quadrotor task for the next version of the paper.
>
>
> | **Methods**        | **Success Rate (Manipulation)** | **Success Rate (Quadrotor)** |
> |---------------------|----------------------------|----------------------------|
> | BC                 | 0.16 ± 0.15               | 0.05 ± 0.04                |
> | DAgger             | 0.34 ± 0.31               | 0.08 ± 0.03             |
> | **HLRL**               | 0.61 ± 0.22               | 0.31 ± 0.11         |
> | **DWBC**               | 0.63 ± 0.18               | 0.35 ± 0.07       |
> | **COSIL**               | 0.56 ± 0.21               | tbd                    |
> | w/o Align (Ours)   | 0.61 ± 0.18               | 0.38 ± 0.11          |
> | w Align (Ours)     | **0.88 ± 0.07**           | **0.46 ± 0.04**    |
> #### **Updated baseline experiments.**

---

### Author Response · Authors · 2024-12-04
**Final Post-Rebuttal Response**

Once again, we would like to thank all reviewers and area chairs for their constructive feedback and discussions. We greatly value the reviewers' thoughtful feedback on clarity, related work, and additional experiments. Based on this helpful feedback and the ensuing discussions, we have significantly improved the manuscript. This improvement is reflected in the scores raised to positive levels by three out of four reviewers. We kindly hope the fourth reviewer may find our responses and enhancements sufficient to reconsider their score as we believe to have adequately addressed their concerns.
The improvements to the manuscript are summarized below:

- **Ablations**: We conducted ablations to evaluate the impact of different components of our proposed method. These results are now included in the updated manuscript. The results show that our method is robust to hyperparameter choices and that all proposed components are beneficial for significantly improving performance.

- **Related Work**: We have incorporated all the missing references suggested by the reviewers and expanded the discussion to better contextualize our contributions.

- **Baselines**: To more effectively highlight the advantages and performance improvements of our method, we have added the results of three baseline methods (HLRL, COSIL, DWBC). Given the complexity of the tasks, these comparisons further demonstrate the effectiveness of our approach.

- **Technical Clarity**: We enhanced the clarity of the methodology and experiments by restructuring sections, improving descriptions, and providing additional technical details. For the rebuttal phase, we opted for short and concise changes to the manuscript to ease the iteration with the reviewers. We will make sure to further improve the clarity and coherence of the text for the final version.

We would like to highlight that the additional experiments were conducted with due diligence. For each baseline method, we performed a parameter search and reported the mean and variance of the performance of five (vision-based manipulation) and three (vision-based quadrotor flight) seeds. This guarantees a fair comparison. The final performance of most baselines with a fixed teacher converges to a similar performance (significantly lower than our approach), confirming the benefit of changing the teacher during training.

---

**Final Updated Experimental Results**


| **Methods**        | **Success Rate (Manipulation)** | **Success Rate (Quadrotor)** |
|---------------------|----------------------------|----------------------------|
| BC                 | 0.16 ± 0.15               | 0.05 ± 0.04               |
| DAgger             | 0.34 ± 0.31               | 0.08 ± 0.03               |
| **HLRL**               | 0.61 ± 0.22               | 0.31 ±   0.11                   |
| **DWBC**               | 0.63 ± 0.18               | 0.35 ± 0.07               |
| **COSIL**               | 0.56 ± 0.21               | 0.30 ± 0.07               |
| w/o Align (Ours)   | 0.61 ± 0.18               | 0.38 ± 0.11               |
| w Align (Ours)     | **0.88 ± 0.07**           | **0.46 ± 0.04**           |
#### **Updated baseline experiments.**

---


| **Configuration**   | **Success Rate**       |
|-------------------------------|-------------------------|
| w Reward / wo Loss       | 0.47 ± 0.37           |
| wo Reward / w Loss       | 0.74 ± 0.08           |
| wo shared decoder        | 0.62 ± 0.27           |
| λ = 0.1                  | 0.81 ± 0.20           |
| λ = 0.075                | 0.77 ± 0.18           |
| (Default) λ = 0.05       | 0.88 ± 0.07           |
| λ = 0.025                | 0.95 ± 0.03           |0.95 ± 0.03   |
#### **Ablation Experiments.**

---

### Public Comment · ~Philip_Bachman1 · 2025-02-24
**Prior work on student guided teacher/student training**

I actually proposed student guided teacher/student training back in 2015 in a paper called: "Data Generation as Sequential Decision Making". See Section 2.2 on page 2 of https://arxiv.org/abs/1506.03504. I called it Generalized Guided Policy Search, since it's a generalization of Guided Policy Search.

One significant difference between your work and mine is that my experiments were in a simpler setting where the student's observations were a subset of the teacher's observations, so there were no concerns about cost of producing the student's observations. Approximating the student's behavior with an additional policy that predicts the student's actions while conditioning on the teacher's observations is a nice trick for making this setup more effective/efficient in contexts where the student's observations are costly.

---

> ### Public Comment · ~Nico_Messikommer1 · 2025-03-17
> **Response: Prior work on student guided teacher/student training**
>
> Dear Philip Bachman,
> Many thanks for pointing us to your work. We have included your work in the paper.

---

### Meta-Review · Area_Chair_55eR · 2024-12-21

**Metareview:**

This paper presents a teacher-student training framework where the teacher learns to generate data that is easier for the student to mimic. In the asymmetric teacher-student setting, the teacher has privileged sensory information while the student is limited and therefore may not be able to mimic the optimal behavior of the teacher. The proposed method leverages action discrepancy between the student and the teacher as a penalty in training the teacher with reinforcement learning.

Reviewers agree this paper studies an interesting and important problem and the proposed method is novel. The experimental results are intriguing, especially in the case where the robot learned to generate camera view-aware policy. However, many baselines and ablation results were added during the rebuttal phase and the reviewers expressed concerns about the readiness of the presentation of the paper. The authors should incorporate additional feedback from the reviewers.

**Additional Comments On Reviewer Discussion:**

Reviewers raised concerns about the lack of sufficient comparison with baselines and ablation of design choices. The authors ran additional experiments during the rebuttal phase and updated the manuscript accordingly to address such concerns. Reviewers raised scores but expressed concerns that the results are rushed.

---

### Decision · Program_Chairs · 2025-01-22

Accept (Spotlight)